# Optimizing Conditional Value-At-Risk of Black-Box Functions

**Quoc Phong Nguyen, Zhongxiang Dai, Bryan Kian Hsiang Low, and Patrick Jaillet**[†]
Dept. of Computer Science, National University of Singapore, Republic of Singapore
Dept. of Electrical Engineering and Computer Science, MIT, USA[†]
{qphong,daizhongxiang,lowkh}@comp.nus.edu.sg, jaillet@mit.edu[†]

## Abstract

This paper presents two *Bayesian optimization* (BO) algorithms with theoretical performance guarantee to maximize the *conditional value-at-risk* (CVaR) of a black-box function: CV-UCB and CV-TS which are based on the well-established principle of optimism in the face of uncertainty and Thompson sampling, respectively. To achieve this, we develop an upper confidence bound of CVaR and prove the no-regret guarantee of CV-UCB by utilizing an interesting connection between CVaR and *value-at-risk* (VaR). For CV-TS, though it is straightforwardly performed with Thompson sampling, bounding its Bayesian regret is non-trivial because it requires a tail expectation bound for the distribution of CVaR of a black-box function, which has not been shown in the literature. The performances of both CV-UCB and CV-TS are empirically evaluated in optimizing CVaR of synthetic benchmark functions and simulated real-world optimization problems.

## 1 Introduction

A wide range of applications from Auto-ML [15] to chemistry [6] and drug design [3] require optimizing a *black-box* objective function (i.e., its closed-form expression, gradient, and convexity are unknown) through observing noisy function evaluations. To resolve this problem, an efficient class of algorithms, called *Bayesian optimization* (BO) [2, 20], has seen rapid development lately.

One of the recent developments of BO considers searching for the *optimization variable* $\mathbf{x}_*$ that maximizes the objective function $\rho[f(\mathbf{x}, \mathbf{W})]$ [4]. Different from the classical BO, there exists an *environmental random variable* $\mathbf{W}$ that cannot be controlled, which happens frequently in real-world problems. For example, to maximize the crop yield (i.e., $f(\mathbf{x}, \mathbf{W})$) by controlling the amount of fertilizer (i.e., $\mathbf{x}$), there are various uncontrollable weather conditions such as the temperature, lighting, and rainfall (i.e., $\mathbf{W}$) [13]. Thus, even though an amount of fertilizer $\mathbf{x}$ increases the crop yield most of the time, there exists a risk/chance that under certain weather conditions (realizations of $\mathbf{W}$), the same amount of fertilizer leads to a low crop yield (an undesirable realization of the random variable $f(\mathbf{x}, \mathbf{W})$). This issue renders the optimization of $f(\mathbf{x}, \mathbf{W})$ futile, so the work of [4] proposes to optimize a risk measure, denoted as $\rho$, of $f(\mathbf{x}, \mathbf{W})$ such as *value-at-risk* (VaR) and *conditional VaR* (CVaR). However, its approach suffers from two main shortcomings: the computational cost of a nested optimization procedure and the lack of theoretical performance guarantee. Although these drawbacks are addressed in the work of [13], its proposed algorithm only works for optimizing VaR of a black-box function. Hence, an efficient algorithm with theoretical performance guarantee for optimizing CVaR of a black-box function remains an open question.

Nonetheless, optimizing CVaR of a black-box function is desirable, especially when the effect of optimizing CVaR cannot be obtained by optimizing VaR. A prominent property of CVaR is its sensitivity to values at the extreme tails of the random variable $f(\mathbf{x}, \mathbf{W})$, which VaR does not exhibit [19]. Let us consider two portfolio allocation schemes: one scheme has an unacceptable

35th Conference on Neural Information Processing Systems (NeurIPS 2021).

worst-case value of the return and the other does not. It is possible that the two schemes have the same VaR of the return because VaR is insensitive to the extreme tails of the return (e.g., the worst-case value of the return). On the other hand, CVaR is able to distinguish the risks of the two schemes. This advantage of CVaR is further clarified in Remark 1.

In this paper, we adopt the settings of [4, 13]: the distribution of $\mathbf{W}$ is given and we can control/select the realization $\mathbf{w}$ of $\mathbf{W}$ in the optimization procedure. In practice, the distribution of $\mathbf{W}$ can be estimated from data (e.g., weather historical data in the crop yield optimization example) and we can perform BO in a laboratory or using computer simulation where $\mathbf{W}$ is controlled [4, 13]. After the optimization, the optimal $\mathbf{x}_*$ can be used in the real-world environment with the uncontrollable $\mathbf{W}$. Our main contribution is to propose two computationally efficient algorithms (Sec. 3) with theoretical performance guarantee (Sec. 4) for optimizing CVaR of a black-box function.

First, we propose *CVaR optimization with upper confidence bound* (CV-UCB) that is based on the well-established principle of optimism in the face of uncertainty. By exploiting the connection between CVaR and VaR, we are able to develop a confidence bound of CVaR (Sec. 3.1.1) and prove the no-regret guarantee of CV-UCB (Sec. 4.1).

Second, we propose *CVaR optimization with Thompson sampling* (CV-TS) which can be naturally extended to handle batch queries (i.e., gathering observations at a batch of inputs in each BO iteration). The capability of gathering observations in a batch simultaneously is often available and preferable in laboratory settings and computer simulations. Though CV-TS can be simply performed with the popular Thompson sampling (or posterior sampling) [18] (Sec. 3.1.2), bounding its Bayesian regret is challenging as the distribution of CVaR in our problem has not been investigated before and its support is unbounded. Fortunately, we are able to relate the problem of bounding the tail expectation of CVaR to that of the function evaluation $f(\mathbf{x}, \mathbf{w})$ (Sec. 4.2). The latter follows a Gaussian distribution with known tail expectation, unlike that of CVaR.

We empirically evaluate the performance of both CV-UCB and CV-TS in optimizing CVaR of synthetic benchmark functions, an optimization problem of the residuary resistance per unit weight of displacement of a yacht, a portfolio optimization problem, and a simulated robot task (Sec. 5).

**Related works.** Existing solutions to optimizing a black-box function $f(\mathbf{x}, \mathbf{W})$ with an environmental random variable $\mathbf{W}$ are different in the assumptions about $\mathbf{W}$ and the objective function. Other than the aforementioned works of [4, 13] where we adopt the assumptions that the distribution of $\mathbf{W}$ is known and $\mathbf{w}$ can be selected during the optimization procedure, the work of [9] also proposes an efficient algorithm with theoretical performance guarantee under the same assumptions to maximize the *probabilistic threshold robustness measure*. This measure requires a threshold of the desirable performance, instead of a *risk level* (i.e., the probability of undesirable performance) in CVaR and VaR. Another related work is *adversarially robust BO* [1] which maximizes $f(\mathbf{x}, \mathbf{w})$ with the worst-case realization $\mathbf{w}$ of $\mathbf{W}$. It can be cast as an optimization problem of VaR of a black-box function [13]. The work of [10] assumes that the distribution of $\mathbf{W}$ is given, but $\mathbf{W}$ is sampled from its distribution during the optimization. Its objective is to balance between the mean and the variance of $f(\mathbf{x}, \mathbf{W})$ from different perspectives: multi-task learning, multi-objective optimization, and constrained optimization. The work of [22] maximizes VaR and CVaR of a black-box function, yet, it assumes $\mathbf{W}$ is unobservable and its distribution is unknown. It suffers from an approximate posterior belief and the lack of theoretical performance guarantee. The works of [12, 14] (namely, *distributional robust BO*) assume that the unknown distribution of $\mathbf{W}$ belongs to an uncertainty set of distributions. They maximize the mean of $f(\mathbf{x}, \mathbf{W})$ under the worst-case realization of the unknown distribution of $\mathbf{W}$ in the uncertainty set.

## 2 Background

### 2.1 Conditional Value-at-Risk

Let the black-box function of interest be a real-valued function $f(\mathbf{x}, \mathbf{w})$ where $\mathbf{x} \in \mathbb{X} \subset \mathbb{R}^m$ and $\mathbf{w} \in \mathbb{W} \subset \mathbb{R}^n$. Let $\mathbf{W}$ denote the *environmental random variable* whose probability distribution is specified by $p(\mathbf{W} = \mathbf{w})$ for all $\mathbf{w} \in \mathbb{W}$. Let $f(\mathbf{x}, \mathbf{W})$ denote the random variable representing the function evaluation at $\mathbf{x}$ and a random realization of $\mathbf{W}$. The *conditional value-at-risk* (CVaR) of $f(\mathbf{x}, \mathbf{W})$ at *risk level* $\alpha \in (0, 1)$ is defined as:

$$c_f(\mathbf{x}; \alpha) \triangleq \mathbb{E}\left[f(\mathbf{x}, \mathbf{W})| f(\mathbf{x}, \mathbf{W}) \leq v_f(\mathbf{x}; \alpha)\right] \tag{1}$$

which is the expectation of $f(\mathbf{x}, \mathbf{W})$ over function values that are at most the *value-at-risk* (VaR) at the same risk level $\alpha$ defined as $v_f(\mathbf{x}; \alpha) \triangleq \inf\{\omega : P(f(\mathbf{x}, \mathbf{W}) \leq \omega) \geq \alpha\}$.

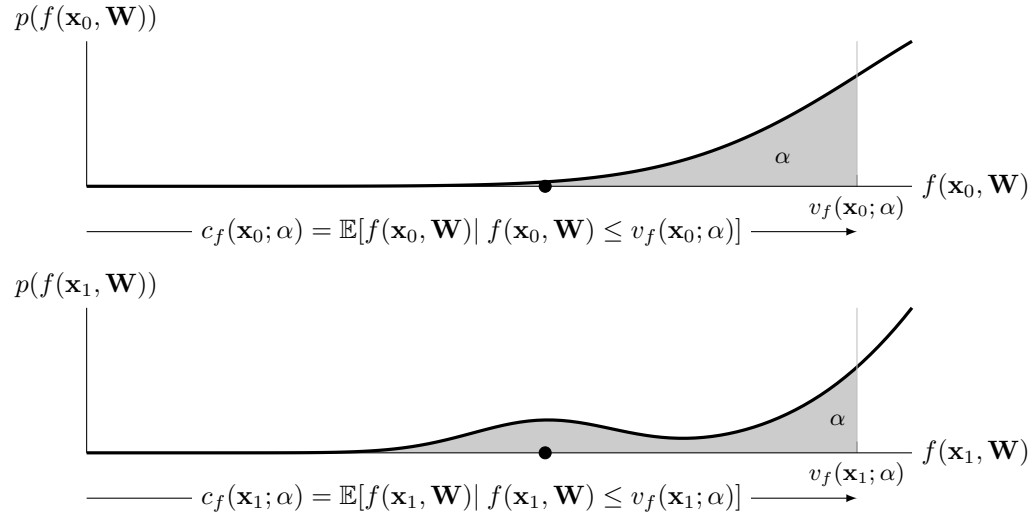

Figure 1: Plot of the lower tails of $f(\mathbf{x}_0, \mathbf{W})$ and $f(\mathbf{x}_1, \mathbf{W})$ with their VaR and CVaR at risk level $\alpha$. The shaded area is $\alpha$. It is observed that $v_f(\mathbf{x}_0; \alpha) = v_f(\mathbf{x}_1; \alpha)$ but $c_f(\mathbf{x}_0; \alpha) > c_f(\mathbf{x}_1; \alpha)$.

**Remark 1** (CVaR vs. VaR). While VaR at $\alpha$ is interpreted as the $\alpha$-percentile of $f(\mathbf{x}, \mathbf{W})$, CVaR is interpreted as the expectation of function values that are at most VaR. Therefore, CVaR is sensitive to the function values less than VaR, while VaR is not. Fig. 1 shows an example of function evaluations at $\mathbf{x}_0$ and $\mathbf{x}_1$ such that their VaR values (i.e., their $\alpha$-percentiles) are the same. We observe that VaR is insensitive to the tails of $f(\mathbf{x}, \mathbf{W})$ which are different in this case (see the region around the black dot). In contrast, $c_f(\mathbf{x}_0; \alpha) > c_f(\mathbf{x}_1; \alpha)$ which conveys the idea that $f(\mathbf{x}_0, \mathbf{W})$ has a lower risk of low function values as the probability density of function values less than VaR concentrates in a region nearer to VaR than that of $f(\mathbf{x}_1, \mathbf{W})$. Thus, CVaR is more risk-aversed than VaR, especially when the distribution of $f(\mathbf{x}, \mathbf{W})$ has a heavy lower tail, which makes it more suitable for critical applications. In the portfolio optimization problem, if the distribution of the portfolio's return has negative outliers (which are captured in CVaR), the average return in the long run can be negative even though the VaR of the return is positive.

In this work, we assume that $|\mathbb{W}|$ is finite to simplify the theoretical analysis (described in Sec. 4) and the evaluation of CVaR. The latter requires the computation of the cumulative probability mass of $f(\mathbf{x}, \mathbf{W})$ by enumerating all possible realizations of $f(\mathbf{x}, \mathbf{W})$ (described in [4]).

### 2.2 Bayesian Optimization of CVaR and Gaussian Processes

**BO of CVaR.** In this paper, we would like to maximize CVaR $c_f(\mathbf{x}; \alpha)$ (1) of $f(\mathbf{x}, \mathbf{W})$ given the distribution $p(\mathbf{W})$ by iteratively gathering observations as noisy function evaluations $y(\mathbf{x}, \mathbf{w}) \triangleq f(\mathbf{x}, \mathbf{w}) + \epsilon(\mathbf{x}, \mathbf{w})$ where $\epsilon(\mathbf{x}, \mathbf{w}) \sim \mathcal{N}(0, \sigma_n^2)$. The gist of BO of CVaR is a strategy to select the *input query* $(\mathbf{x}_t, \mathbf{w}_t)$ at iteration $t$ such that the maximizer $\mathbf{x}_* \in \arg\max_{\mathbf{x} \in \mathbb{X}} c_f(\mathbf{x}; \alpha)$ is found as rapidly as possible. Note that while $\mathbf{W}$ is controllable in BO, after the optimization, the optimal $\mathbf{x}_*$ can be used in real-world settings where $\mathbf{W}$ is uncontrollable. At iteration $t$, we have all observations at input queries in the previous $t - 1$ iterations, denote as $\mathbf{D}_{t-1} = \mathbf{D}_{t-2} \cup \{(\mathbf{x}_{t-1}, \mathbf{w}_{t-1})\}$ ($\mathbf{D}_0$ denotes the set of initial observed inputs). These observations $\mathbf{y}_{\mathbf{D}_{t-1}} \triangleq (y(\mathbf{x}, \mathbf{w}))_{(\mathbf{x}, \mathbf{w}) \in \mathbf{D}_{t-1}}$ are used to construct the *posterior* belief/distribution of the underlying black-box function $f(\mathbf{x}, \mathbf{w})$ which is utilized in building the query selection strategy. Let us revisit Fig. 1: $c_f(\mathbf{x}_0; \alpha) > c_f(\mathbf{x}_1; \alpha)$ implies that $\mathbf{x}_0$ is preferred to $\mathbf{x}_1$. It is because the risk/chance of getting low function values at $\mathbf{x}_0$ is smaller (see the region around the black dot). In contrast, VaR cannot differentiate the risk of $f(\mathbf{x}_0, \mathbf{W})$ and that of $f(\mathbf{x}_1, \mathbf{W})$ since their VaR values are the same.

---

**Algorithm 1** BO Algorithms for optimizing CVaR of a black-box function

---

1: **Input:** algo, $\mathbb{X}$, $\mathbb{W}$, initial observation $\mathbf{y}_{\mathbf{D}_0}$, prior $\mu_0 = 0, \sigma_n, \kappa$
2: **for** $t = 1, 2, \ldots$ **do**
3:     {*Selecting* $\mathbf{x}_t$}
4:     **if** algo is CV-UCB **then**
5:         Select $\mathbf{x}_t \in \underset{\mathbf{x}}{\operatorname{argmax}}\, c_{u_{t-1}}(\mathbf{x}; \alpha)$
6:     **else if** algo is CV-TS **then**
7:         Sample a function $f_1$ from the GP posterior belief given $\mathbf{y}_{\mathbf{D}_{t-1}}$
8:         Select $\mathbf{x}_t \in \underset{\mathbf{x}}{\operatorname{argmax}}\, c_{f_1}(\mathbf{x}; \alpha)$
9:     **end if**
10:    {*Selecting* $\mathbf{w}_t$}
11:    Find $\alpha_t \in \arg \underset{\alpha' \in (0, \alpha]}{\max}\, v_{u_{t-1}}(\mathbf{x}_t; \alpha') - v_{l_{t-1}}(\mathbf{x}_t; \alpha')$
12:    Given $\alpha_t$, select $\mathbf{w}_t$ as an LV w.r.t. $\mathbf{x}_t$, $u_{t-1}$, and $l_{t-1}$
13:    {*Collecting data and updating GP*}
14:    Incorporate new observation at input query: $\mathbf{y}_{\mathbf{D}_t} = \mathbf{y}_{\mathbf{D}_{t-1}} \cup \{y(\mathbf{x}_t, \mathbf{w}_t)\}$
15:    Update the GP posterior belief given $\mathbf{y}_{\mathbf{D}_t}$
16: **end for**

---

**Gaussian process (GP).** In order to obtain the posterior belief of $f(\mathbf{x}, \mathbf{w})$ given noisy observations $\mathbf{y}_{\mathbf{D}_{t-1}}$, we model the black-box function $f$ with a GP: every finite subset of $\{f(\mathbf{x}, \mathbf{w})\}_{(\mathbf{x}, \mathbf{w}) \in \mathbb{X} \times \mathbb{W}}$ follows a multivariate Gaussian distribution [17]. Thus, GP is fully specified by its *prior* mean which is assumed to be zero and its *covariance function/kernel* $\kappa(\mathbf{x}, \mathbf{w}; \mathbf{x}', \mathbf{w}') \triangleq cov[f(\mathbf{x}, \mathbf{w}), f(\mathbf{x}', \mathbf{w}')]$. Given noisy observations $\mathbf{y}_{\mathbf{D}_{t-1}}$ at iteration $t$, the posterior belief $p(f(\mathbf{x}, \mathbf{w}) | \mathbf{y}_{\mathbf{D}_{t-1}})$ is a Gaussian specified by the following mean and variance:

$$
\begin{aligned}
\mu_{t-1}(\mathbf{x}, \mathbf{w}) &\triangleq \mathbf{K}_{\mathbf{x}, \mathbf{w}; \mathbf{D}_{t-1}} \mathbf{\Lambda}_{\mathbf{D}_{t-1}} \mathbf{y}_{\mathbf{D}_{t-1}} \\
\sigma_{t-1}^2(\mathbf{x}, \mathbf{w}) &\triangleq \kappa(\mathbf{x}, \mathbf{w}) - \mathbf{K}_{\mathbf{x}, \mathbf{w}; \mathbf{D}_{t-1}} \mathbf{\Lambda}_{\mathbf{D}_{t-1}} \mathbf{K}_{\mathbf{D}_{t-1}; \mathbf{x}, \mathbf{w}}
\end{aligned}
\tag{2}
$$

where $\kappa(\mathbf{x}, \mathbf{w}) \triangleq \kappa(\mathbf{x}, \mathbf{w}; \mathbf{x}, \mathbf{w})$, $\mathbf{K}_{\mathbf{x}, \mathbf{w}; \mathbf{D}_{t-1}} \triangleq (\kappa(\mathbf{x}, \mathbf{w}; \mathbf{x}', \mathbf{w}'))_{(\mathbf{x}', \mathbf{w}') \in \mathbf{D}_{t-1}}$, $\mathbf{K}_{\mathbf{D}_{t-1}; \mathbf{x}, \mathbf{w}} \triangleq \mathbf{K}_{\mathbf{x}, \mathbf{w}; \mathbf{D}_{t-1}}^{\top}$, and $\mathbf{\Lambda}_{\mathbf{D}_{t-1}} \triangleq (\mathbf{K}_{\mathbf{D}_{t-1} \mathbf{D}_{t-1}} + \sigma_n^2 \mathbf{I})^{-1}$ ($\mathbf{I}$ is the identity matrix and $\mathbf{K}_{\mathbf{D}_{t-1} \mathbf{D}_{t-1}} \triangleq (\kappa(\mathbf{x}', \mathbf{w}'; \mathbf{x}'', \mathbf{w}''))_{(\mathbf{x}', \mathbf{w}', \mathbf{x}'', \mathbf{w}'') \in \mathbf{D}_{t-1} \times \mathbf{D}_{t-1}}$).

Although $f(\mathbf{x}, \mathbf{w})$ follows a Gaussian distribution, CVaR $c_f(\mathbf{x}; \alpha)$ (defined in (1)) does not. Furthermore, although $|\mathbb{W}|$ is finite, the distribution of CVaR is continuous because the black-box function evaluation is real-valued. Besides, as the support of $f(\mathbf{x}, \mathbf{W})$ is unbounded, so is CVaR. These points pose challenges in deriving the following 2 algorithms and their theoretical analyses.

## 3 BO Algorithms for Optimizing CVaR of a Black-Box Function

In this section, we present two BO algorithms for optimizing CVaR of a black-box function $f(\mathbf{x}, \mathbf{W})$: *CVaR optimization with upper confidence bound* (CV-UCB) and *CVaR optimization with Thompson sampling* (CV-TS). These two algorithms are different in the way of handling the exploration-exploitation trade-off in the selection of $\mathbf{x}_t$ to find $\operatorname{argmax}_{\mathbf{x} \in \mathbb{X}} c_f(\mathbf{x}; \alpha)$ (Sec. 3.1.1 and Sec. 3.1.2). On the other hand, given the selected $\mathbf{x}_t$, selecting $\mathbf{w}_t$ is only about reducing the uncertainty of CVaR $c_f(\mathbf{x}_t; \alpha)$. Therefore, we would like to design a single selection strategy of $\mathbf{w}_t$ that works for both CV-UCB and CV-TS (Sec. 3.2).

### 3.1 Selection Strategy of $\mathbf{x}_t$

#### 3.1.1 A UCB-based Approach to Select $\mathbf{x}_t$ in CV-UCB

In (1), CVaR $c_f(\mathbf{x}; \alpha)$ at risk level $\alpha$ of $f(\mathbf{x}, \mathbf{W})$ is defined as the expectation of function values that are at most VaR at risk level $\alpha$ [4]. While this definition (1) is interpretable, it is challenging to devise a no-regret BO algorithm based on (1). It is because (1) requires learning about not only (i) VaR $v_f(\mathbf{x}; \alpha)$ but also (ii) the probabilities of function values less than $v_f(\mathbf{x}; \alpha)$. Even though the former (i.e., VaR $v_f(\mathbf{x}; \alpha)$) can be optimized using the V-UCB algorithm in [13], it remains a challenge to learn about the latter.

Interestingly, in this section, we present an approach that does not resort to handling the above two problems (i) and (ii) separately. To achieve this, we employ an alternative definition of CVaR which is an expectation of VaR over the risk level:

$$c_f(\mathbf{x};\alpha) = \frac{1}{\alpha}\int_0^\alpha v_f(\mathbf{x};\alpha')\,\mathrm{d}\alpha'\ . \tag{3}$$

Although (1) is more interpretable and is often used to evaluate CVaR, the above definition (3) paves the way to our selection strategy of $\mathbf{x}_t$ in CV-UCB: by utilizing (3), we are able to construct an upper confidence bound of CVaR relying on that of VaR. Recall that the work of [13] proposes a confidence bound of VaR at iteration $t$, denoted as $I_{t-1}[v_f(\mathbf{x};\alpha')]$:

$$I_{t-1}[v_f(\mathbf{x};\alpha')] \triangleq [v_{l_{t-1}}(\mathbf{x};\alpha'), v_{u_{t-1}}(\mathbf{x};\alpha')] \quad \forall \mathbf{x} \in \mathbb{X}\ \forall \alpha' \in (0,1) \tag{4}$$

where $v_{l_{t-1}}(\mathbf{x};\alpha')$ and $v_{u_{t-1}}(\mathbf{x};\alpha')$ are VaR (due to the randomness in $\mathbf{W}$) of $l_{t-1}(\mathbf{x},\mathbf{W})$ and $u_{t-1}(\mathbf{x},\mathbf{W})$, respectively. At iteration $t$, for all $(\mathbf{x},\mathbf{w}) \in \mathbb{X} \times \mathbb{W}$, $[l_{t-1}(\mathbf{x},\mathbf{w}), u_{t-1}(\mathbf{x},\mathbf{w})]$ is a confidence bound of $f(\mathbf{x},\mathbf{w})$ (due to the uncertainty in the unknown black-box function $f$) depending on an *exploration parameter* $\beta_t$ [13]:

$$\begin{aligned} l_{t-1}(\mathbf{x},\mathbf{w}) &\triangleq \mu_{t-1}(\mathbf{x},\mathbf{w}) - \beta_t^{1/2}\sigma_{t-1}(\mathbf{x},\mathbf{w}) \\ u_{t-1}(\mathbf{x},\mathbf{w}) &\triangleq \mu_{t-1}(\mathbf{x},\mathbf{w}) + \beta_t^{1/2}\sigma_{t-1}(\mathbf{x},\mathbf{w}) \end{aligned} \tag{5}$$

where $\mu_{t-1}(\mathbf{x},\mathbf{w})$ and $\sigma_{t-1}(\mathbf{x},\mathbf{w})$ are defined in (2). The exploration parameter $\beta_t$ is to balance between exploitation (based on the GP posterior mean $\mu_{t-1}(\mathbf{x},\mathbf{w})$) and exploration (based on the GP posterior standard deviation $\sigma_{t-1}(\mathbf{x},\mathbf{w})$), which is specified later in Lemma 1. From (3), a confidence bound $I_{t-1}[c_f(\mathbf{x};\alpha)]$ of CVaR follows the confidence bound $I_{t-1}[v_f(\mathbf{x};\alpha')]$ of VaR (4):

$$I_{t-1}[c_f(\mathbf{x};\alpha)] \triangleq [c_{l_{t-1}}(\mathbf{x};\alpha), c_{u_{t-1}}(\mathbf{x};\alpha)] \triangleq \left[\frac{1}{\alpha}\int_0^\alpha v_{l_{t-1}}(\mathbf{x};\alpha')\,\mathrm{d}\alpha', \frac{1}{\alpha}\int_0^\alpha v_{u_{t-1}}(\mathbf{x};\alpha')\,\mathrm{d}\alpha'\right]\ . \tag{6}$$

Given the above upper confidence bound $c_{u_{t-1}}(\mathbf{x};\alpha)$ of CVaR, we select $\mathbf{x}_t$ as its maximizer (line 5 of Algorithm 1). The time complexity of the selection strategy of $\mathbf{x}_t$ in CV-UCB is $\mathcal{O}(|\mathbf{D}_{t-1}|^3 + n_{\text{train}}(|\mathbb{W}||\mathbf{D}_{t-1}|^2 + |\mathbb{W}|\log|\mathbb{W}|))$ where $n_{\text{train}}$ is the number of gradient descent steps to find $\operatorname{argmax}_\mathbf{x} c_{u_{t-1}}(\mathbf{x};\alpha)$, $\mathcal{O}(|\mathbf{D}_{t-1}|^3)$ is to compute $\Lambda_{\mathbf{D}_{t-1}}$ in (2), $\mathcal{O}(|\mathbb{W}||\mathbf{D}_{t-1}|^2)$ is the GP prediction time complexity, and $\mathcal{O}(|\mathbb{W}|\log|\mathbb{W}|)$ is to evaluate CVaR.

### 3.1.2 A Thompson Sampling Approach to Select $\mathbf{x}_t$ in CV-TS

An alternative approach to select $\mathbf{x}_t$ is based on Thompson sampling [18]: $\mathbf{x}_t$ is selected as a sample of the maximizer $\mathbf{x}_*$ of CVaR $c_f(\mathbf{x};\alpha)$. Unlike the selection strategy of $\mathbf{x}_t$ in CV-UCB that requires an exploration parameter $\beta_t$ (Sec. 3.1.1), the exploration-exploitation trade-off in Thompson sampling is naturally handled by the randomness in the sampling of the maximizer of $c_f(\mathbf{x};\alpha)$. This approach can also be extended to handle a batch query at each BO iteration by drawing a batch of samples of the maximizer of $c_f(\mathbf{x};\alpha)$. While unexplored for BO of CVaR, batch queries are popular in BO literature [8, 11].

A sample of the maximizer of $c_f(\mathbf{x};\alpha)$ is obtained by the following 2 steps (lines 7-8 of Algorithm 1). First, we draw a function sample $f_1$ from the GP posterior belief by using random Fourier feature approximation [16] which is also employed in classical BO works [7, 23] (line 7 of Algorithm 1). Second, given $f_1$, we maximize CVaR $c_{f_1}(\mathbf{x};\alpha)$ to obtain its maximizer. This is a sample of the maximizer of $c_f(\mathbf{x};\alpha)$ which is assigned to $\mathbf{x}_t$ in CV-TS (line 8 of Algorithm 1). The time complexity of the selection strategy of $\mathbf{x}_t$ in CV-TS is $\mathcal{O}(|\mathbf{D}_{t-1}|^3 + n_{\text{train}}(n_{\text{feature}}^2 + |\mathbb{W}|\log|\mathbb{W}|))$ where $n_{\text{feature}}$ is the number of random Fourier samples, $n_{\text{train}}$ is the number of gradient descent steps to find $\operatorname{argmax}_\mathbf{x} c_{f_1}(\mathbf{x};\alpha)$, $\mathcal{O}(|\mathbf{D}_{t-1}|^3)$ is to compute $\Lambda_{\mathbf{D}_{t-1}}$ in (2) (which dominates the time complexity to draw a function $f_1$ from the GP posterior belief), $\mathcal{O}(n_{\text{feature}}^2)$ is the time complexity to evaluate $f_1$, and $\mathcal{O}(|\mathbb{W}|\log|\mathbb{W}|)$ is to evaluate CVaR.

### 3.2 Selection Strategy of $\mathbf{w}_t$ Shared by Both CV-UCB & CV-TS

Given the selected $\mathbf{x}_t$ at iteration $t$, we would like to propose a single strategy of selecting $\mathbf{w}_t$ to reduce the uncertainty of CVaR $c_f(\mathbf{x}_t;\alpha)$ for both CV-UCB and CV-TS that still guarantees no-regret for both algorithms as shown in Sec. 4.

Our main idea is to prioritize learning about VaR at risk level $\alpha_t$ which has the *highest uncertainty* among VaR $v_f(\mathbf{x}; \alpha')$ at risk levels $\alpha' \in (0, \alpha]$. In turn, it reduces the uncertainty of CVaR from (3). We quantify the uncertainty of VaR by the size of its confidence bound (4). Thus, $\alpha_t$ is defined as (line 11 of Algorithm 1)

$$\alpha_t \in \arg \max_{\alpha' \in (0, \alpha]} v_{u_{t-1}}(\mathbf{x}_t; \alpha') - v_{l_{t-1}}(\mathbf{x}_t; \alpha') . \tag{7}$$

To reduce the uncertainty of $v_f(\mathbf{x}_t; \alpha_t)$ quantified by the size of its confidence bound $v_{u_{t-1}}(\mathbf{x}_t; \alpha_t) - v_{l_{t-1}}(\mathbf{x}_t; \alpha_t)$, let us view $v_f(\mathbf{x}_t; \alpha_t)$ as the $\alpha_t$-percentile of $f(\mathbf{x}_t, \mathbf{W})$. There is no uncertainty in the $\alpha_t$-percentile of $f(\mathbf{x}_t, \mathbf{W})$ if the set of $\mathbf{w}$ where $f(\mathbf{x}_t, \mathbf{w})$ is at most the $\alpha_t$-quantile of $f(\mathbf{x}_t, \mathbf{W})$ (i.e., the lower tail of $f(\mathbf{x}_t, \mathbf{W})$ upper bounded by its $\alpha_t$-percentile) remains unchanged under different realizations of $f$, especially its lower confidence bound $l_{t-1}$ and its upper confidence bound $u_{t-1}$. On the contrary, if there exists $\mathbf{w}_{\text{LV}}$ that violates the above statement due to the uncertainty of $f(\mathbf{x}_t, \mathbf{w}_{\text{LV}})$ (due to the unknown $f$), we would like to gather observation at $(\mathbf{x}_t, \mathbf{w}_{\text{LV}})$ to reduce the uncertainty of its function evaluation. Such $\mathbf{w}_{\text{LV}}$ is formally defined as the *lacing value* (LV) which is shown to exist in [13].[1]

**Definition 1** (Lacing values [13]). *Lacing values* (LV) with respect to $\alpha \in (0, 1)$, $\mathbf{x} \in \mathbb{X}$, $l_{t-1}$, and $u_{t-1}$ are $\mathbf{w}_{\text{LV}} \in \mathbb{W}$ that satisfies $l_{t-1}(\mathbf{x}, \mathbf{w}_{\text{LV}}) \leq v_{l_{t-1}}(\mathbf{x}; \alpha) \leq v_{u_{t-1}}(\mathbf{x}; \alpha) \leq u_{t-1}(\mathbf{x}, \mathbf{w}_{\text{LV}})$, equivalently, $I_{t-1}[v_f(\mathbf{x}; \alpha)] \subset [l_{t-1}(\mathbf{x}, \mathbf{w}_{\text{LV}}), u_{t-1}(\mathbf{x}, \mathbf{w}_{\text{LV}})]$.

Thus, we select $\mathbf{w}_t$ as an LV w.r.t. $\alpha_t$, $\mathbf{x}_t$, $l_{t-1}$, and $u_{t-1}$. If there are multiple LVs, we select the LV with the maximum probability $p(\mathbf{W})$. It is a heuristic to improve the performance suggested by [13]. For CV-TS, to avoid repeating inputs in the batch query, we can select $\mathbf{w}_t$ as a sample of the random variable $\mathbf{W}$ by restricting its support to the LVs (if there are multiple LVs).

## 4 Theoretical Analysis

This section presents our theoretical analyses of CV-UCB and CV-TS. We choose the exploration parameter $\beta_t$ in (5) following a lemma in [21]. We assume a discrete domain $\mathbb{X}$ for simplicity (and $|\mathbb{W}|$ is finite).

**Lemma 1** (Lemma 5.1 in [21]). Pick $\delta \in (0, 1)$ and set $\beta_t = 2 \log(|\mathbb{X}||\mathbb{W}|\pi_t/\delta)$, where $\sum_{t \geq 1} \pi_t^{-1} = 1$, $\pi_t > 0$. Then,

$$|f(\mathbf{x}, \mathbf{w}) - \mu_{t-1}(\mathbf{x}, \mathbf{w})| \leq \beta_t^{1/2} \sigma_{t-1}(\mathbf{x}, \mathbf{w}) \quad \forall (\mathbf{x}, \mathbf{w}) \in \mathbb{X} \times \mathbb{W} \, \forall t \geq 1 \tag{8}$$

holds with probability $\geq 1 - \delta$.

Thus, given the above $\beta_t$, the function evaluation, VaR, and CVaR are in their confidence bounds (in (5), (4), and (6), respectively) with probability $\geq 1 - \delta$.

### 4.1 CV-UCB

The performance of CV-UCB is measured by the *cumulative regret* adopted from existing BO works, e.g., in [21, 13]: $R_T \triangleq \sum_{t=1}^T r_t$ which is the sum over *instantaneous regrets*, denoted as $r_t \triangleq c_f(\mathbf{x}_*; \alpha) - c_f(\mathbf{x}_t; \alpha)$ where $\mathbf{x}_* \in \arg\max_{\mathbf{x} \in \mathbb{X}} c_f(\mathbf{x}; \alpha)$ is the optimal value of the optimization variable. In this section, we would like to show that the cumulative regret of CV-UCB is sublinear so that $\lim_{T \to \infty} R_T/T = 0$.

Since CV-UCB selects $\mathbf{x}_t$ as the maximizer of the upper confidence bound of CVaR (line 5 of Algorithm 1), it can be shown in Appendix A that the instantaneous regret is bounded by

$$r_t \leq c_{u_{t-1}}(\mathbf{x}_t; \alpha) - c_{l_{t-1}}(\mathbf{x}_t; \alpha) \tag{9}$$

with probability $\geq 1 - \delta$. Then, by exploiting the relationship between CVaR and VaR in (3) and the fact that the average of a set of values is at most the maximum value of the set, the bound on $r_t$ can be simplified to the size of a confidence bound of VaR at risk level $\alpha_t$ (7) in Appendix A:

$$r_t \leq v_{u_{t-1}}(\mathbf{x}_t; \alpha_t) - v_{l_{t-1}}(\mathbf{x}_t; \alpha_t) \tag{10}$$

---

[1]Shrewd readers may notice a difference between the intuition and definition of LV ($\leq$ vs. $<$). It is to handle the case when there is only 1 LV.

holds with probability $\geq 1 - \delta$. Furthermore, utilizing the property of $\mathbf{w}_t$ as an LV in Definition 1, it can be shown in Appendix A that

$$r_t \leq 2\beta_t^{1/2}\sigma_{t-1}(\mathbf{x}_t, \mathbf{w}_t) \tag{11}$$

with probability $\geq 1 - \delta$. Then, we follow [21] to construct the following theorem which shows a sublinear regret bound for CV-UCB (detailed in Appendix B).

**Theorem 1.** By selecting $\mathbf{x}_t$ as the maximizer of the upper confidence bound $c_{u_{t-1}}(\mathbf{x}; \alpha)$ (line 5 of Algorithm 1) and selecting $\mathbf{w}_t$ as an LV w.r.t. $\alpha_t$ (7), $\mathbf{x}_t$, $l_{t-1}$, and $u_{t-1}$ (line 12 of Algorithm 1),

$$R_T \leq \sqrt{C_1 T \beta_T \gamma_T} \tag{12}$$

holds with probability $\geq 1 - \delta$ where $C_1 \triangleq 8/\log(1 + \sigma_n^{-2})$, $\gamma_T$ is the maximum information gain about $f$ that can be obtained by observing any set of $T$ observations (which is bounded in [21] for several commonly used kernels), $\beta_T$ and $\delta$ are defined in Lemma 1.

## 4.2 CV-TS

The performance of CV-TS is measured by the *Bayesian cumulative regret* [18]: $R_T^{\text{Bayes}} \triangleq \mathbb{E}\left[\sum_{t=1}^{T} c_f(\mathbf{x}_*; \alpha) - c_f(\mathbf{x}_t; \alpha)\right]$ where the expectation includes the randomness in the GP prior of $f$, the observation noise, and $\mathbf{x}_t$. Let us denote $r_t^{\text{Bayes}} \triangleq \mathbb{E}[c_f(\mathbf{x}_*; \alpha) - c_f(\mathbf{x}_t; \alpha)|\mathbf{y}_{\mathbf{D}_{t-1}}]$ where the expectation includes the randomness in the GP posterior of $f$ given $\mathbf{y}_{\mathbf{D}_{t-1}}$ and $\mathbf{x}_t$, then $R_T^{\text{Bayes}} = \mathbb{E}\left[\sum_{t-1}^{T} r_t^{\text{Bayes}}\right]$. In this section, we would like to show that the Bayesian cumulative regret of CV-TS is sublinear so that $\lim_{T\to\infty} R_T^{\text{Bayes}}/T = 0$. Following [18], we can decompose $r_t^{\text{Bayes}}$ into (to ease the notation, we omit $\mathbf{y}_{\mathbf{D}_{t-1}}$):

$$r_t^{\text{Bayes}} \leq \mathbb{E}[\Delta_c^{\text{lower}}(\mathbf{x}_t; \alpha)] + \mathbb{E}[\Delta_c^{\text{upper}}(\mathbf{x}_*; \alpha)] + \mathbb{E}[c_{u_{t-1}}(\mathbf{x}_t; \alpha) - c_{l_{t-1}}(\mathbf{x}_t; \alpha)] \tag{13}$$

where $\Delta_c^{\text{lower}}(\mathbf{x}_t; \alpha) \triangleq \max(0, c_{l_{t-1}}(\mathbf{x}_t; \alpha) - c_f(\mathbf{x}_t; \alpha))$ and $\Delta_c^{\text{upper}}(\mathbf{x}_*; \alpha) \triangleq \max(0, c_f(\mathbf{x}_*; \alpha) - c_{u_{t-1}}(\mathbf{x}_*; \alpha))$ (shown in Appendix C).

From Appendix A, $c_{u_{t-1}}(\mathbf{x}_t; \alpha) - c_{l_{t-1}}(\mathbf{x}_t; \alpha)$ is bounded by $2\beta_t^{1/2}\sigma_{t-1}(\mathbf{x}_t, \mathbf{w}_t)$ provided that $\mathbf{w}_t$ is an LV w.r.t. $\alpha_t$ (7), $\mathbf{x}_t$, $l_{t-1}$, and $u_{t-1}$ (line 12 of Algorithm 1). Thus, we are able to obtain a bound on the last term of the decomposed $r_t^{\text{Bayes}}$ (13):

$$\mathbb{E}[c_{u_{t-1}}(\mathbf{x}_t; \alpha) - c_{l_{t-1}}(\mathbf{x}_t; \alpha)] \leq 2\beta_t^{1/2}\sigma_{t-1}(\mathbf{x}_t, \mathbf{w}_t) \, . \tag{14}$$

However, the challenge remains in bounding the first two terms (i.e., $\mathbb{E}[\Delta_c^{\text{lower}}(\mathbf{x}_t; \alpha)]$ and $\mathbb{E}[\Delta_c^{\text{upper}}(\mathbf{x}_*; \alpha)]$) of the decomposed $r_t^{\text{Bayes}}$ (13). These terms are the expectations over the tails of CVaR. As suggested by the seminal work of [18], bounding these terms may require CVaR to have a bounded support or a strong tail decay. However, in our BO formulation where $f$ is modelled with a GP, the support of CVaR is unbounded. Thus, to derive the bound of the tail expectation of the unfamiliar distribution of CVaR, our key idea is to relate it to the tail expectation of a Gaussian distribution through 3 steps: **first**, relating this tail expectation of CVaR to that of VaR in Lemma 2, **second**, relating the tail expectation of VaR to the tail probability of VaR in (16), and **third**, relating the tail probability of VaR to that of function evaluation $f(\mathbf{x}, \mathbf{w})$ in Lemma 3. As $f(\mathbf{x}, \mathbf{w})$ follows a Gaussian distribution, we can bound its tail expectation.

**Lemma 2** (From tail expectation of CVaR to that of VaR)**.** For all $\mathbf{x} \in \mathbb{X}$,

$$\mathbb{E}\left[\Delta_c^{\text{lower}}(\mathbf{x}; \alpha)\right] \leq \frac{1}{\alpha} \int_0^{\alpha} \mathbb{E}\left[\Delta_v^{\text{lower}}(\mathbf{x}; \alpha')\right] \, \mathrm{d}\alpha' \tag{15}$$

where $\Delta_v^{\text{lower}}(\mathbf{x}; \alpha') \triangleq \max\left(0, v_{l_{t-1}}(\mathbf{x}; \alpha') - v_f(\mathbf{x}; \alpha')\right)$ (shown in Appendix D).

Furthermore, as $\Delta_v^{\text{lower}}(\mathbf{x}; \alpha')$ is a non-negative random variable, its expectation in (15) can be expressed as an integration over the tail probabilities: $\forall \alpha' \in (0, 1)$, $\mathbf{x} \in \mathbb{X}$,

$$\mathbb{E}\left[\Delta_v^{\text{lower}}(\mathbf{x}; \alpha')\right] = \int_0^{\infty} P(\Delta_v^{\text{lower}}(\mathbf{x}; \alpha') > \omega) \, \mathrm{d}\omega = \int_0^{\infty} P(v_{l_{t-1}}(\mathbf{x}; \alpha') - v_f(\mathbf{x}; \alpha') > \omega) \, \mathrm{d}\omega \, . \tag{16}$$

Then, we are able to relate the tail probability $P(v_{l_{t-1}}(\mathbf{x}; \alpha') - v_f(\mathbf{x}; \alpha') > \omega)$ to the tail probability of the function evaluation $f(\mathbf{x}, \mathbf{w})$ using our key observation on the relationship between the 2 events: (a) VaR $v_f(\mathbf{x}; \alpha')$ falls below its lower confidence bound $v_{l_{t-1}}(\mathbf{x}; \alpha')$ by $\omega \geq 0$ and (b) the function evaluation $f(\mathbf{x}, \mathbf{w})$ falls below its lower confidence bound $l_{t-1}(\mathbf{x}, \mathbf{w})$ by $\omega$ in the following lemma (proved in Appendix E).

**Lemma 3.** Consider a realization $f_1$ of the black-box function $f$ following the GP posterior belief given $\mathbf{y}_{\mathbf{D}_{t-1}}$ that satisfies

$$v_{l_{t-1}}(\mathbf{x}; \alpha') - v_{f_1}(\mathbf{x}; \alpha') > \omega$$

for $\alpha' \in (0, 1)$, $\mathbf{x} \in \mathbb{X}$, and $\omega \geq 0$. Let $\mathbb{W}^{\mathrm{upper}}_{l_{t-1}} \triangleq \{\mathbf{w} \in \mathbb{W} : l_{t-1}(\mathbf{x}, \mathbf{w}) \geq v_{l_{t-1}}(\mathbf{x}; \alpha')\}$. Then,

$$\exists \mathbf{w}_0 \in \mathbb{W}^{\mathrm{upper}}_{l_{t-1}}, \; l_{t-1}(\mathbf{x}, \mathbf{w}_0) - f_1(\mathbf{x}, \mathbf{w}_0) > \omega \;.$$

As the tail expectation of the Gaussian random variable $f(\mathbf{x}, \mathbf{w})$ can be evaluated, we can utilize the above results ((15), (16), and Lemma 3) to obtain a bound of $\mathbb{E}[\Delta_c^{\mathrm{lower}}(\mathbf{x}; \alpha)]$ (and similarly, a bound of $\mathbb{E}[\Delta_c^{\mathrm{upper}}(\mathbf{x}; \alpha)]$) for all $\mathbf{x} \in \mathbb{X}$. These bounds and (14) lead to the following theorem on the Bayesian cumulative regret bound (proved in Appendix F).

**Theorem 2.** Assuming a bounded GP prior standard deviation: $\kappa(\mathbf{x}, \mathbf{w}) \leq 1 \; \forall (\mathbf{x}, \mathbf{w}) \in \mathbb{X} \times \mathbb{W}$, by selecting $\mathbf{x}_t$ as a sample of the maximizer of $c_f(\mathbf{x}; \alpha)$ (lines 7-8 of Algorithm 1) and selecting $\mathbf{w}_t$ as an LV w.r.t. $\alpha_t$ (7), $\mathbf{x}_t$, $l_{t-1}$, and $u_{t-1}$ (line 12 of Algorithm 1), then the Bayesian cumulative regret is bounded by

$$R_T^{\mathrm{Bayes}} \leq \frac{\delta\sqrt{2}}{|\mathbb{X}|\sqrt{\pi}} + \sqrt{C_1 T \beta_T \gamma_T} \tag{17}$$

where $C_1 \triangleq 8/\log(1 + \sigma_n^{-2})$, $\gamma_T$ is the maximum information gain about $f$ that can be obtained by observing any set of $T$ observations, $\beta_T$ and $\delta$ are defined in Lemma 1.

**Remark 2.** We can extend CV-TS to select a batch query of size $k$ (instead of a single query) at each BO iteration by drawing $k$ samples of the maximizer of CVaR and finding their corresponding $\alpha_t$ and LVs (detailed in Appendix G). We can also show that the difference between the average (over the number of observations) of the Bayesian cumulative regret bound of CV-TS with batch queries and that of CV-TS with single queries is small for a large number of observations (Appendix G), which is similar to a result in [11] for the classical BO with Thompson sampling. However, with a larger batch size $k$, more observations can be obtained given the same number of BO iterations. Thus, the use of CV-TS with a large batch size $k$ is advantageous if obtaining observations at the batch query is feasible, e.g., by running multiple simulations in parallel.

**Remark 3.** Our CV-TS algorithm and its theoretical analysis can be utilized to construct a BO algorithm to optimize VaR of a black-box function. Like CV-TS, this algorithm is also capable of handling batch queries. Since it is beyond the scope of this work (which is about CVaR), we refer readers interested in optimizing VaR of a black-box function to Appendix H.

## 5   Experiments

We empirically evaluate the performance of CV-UCB and CV-TS by comparing them with the work of [4]. To the best of our knowledge, it is the only existing work that optimizes CVaR of a black-box function by selecting both $\mathbf{x}$ and $\mathbf{w}$. In particular, the $\rho\mathrm{KG}^{apx}$ algorithm is selected as a baseline thanks to its time efficiency and competitive empirical performance as demonstrated in [4]. There are 2 variants of CV-TS in the experiments: *CV-TS $k = 1$* which selects 1 input query at each iteration and *CV-TS $k = 3$* which selects 3 input queries at each iteration. Each experiment is repeated 10 times with different random seeds to account for the randomness in the observation noise, the set of initial observations, and the sampling from the GP posterior belief. Both the average and the confidence interval of the logarithm of the inference regret (taken from [7, 23]) are reported. The hyperparameters of GP and the noise variance are learned from the observations using maximum likelihood estimation [17]. Further details on the experiments are described in Appendix I.[2]

Figs. 2a-d show the results of optimizing CVaR of synthetic benchmark functions with low input dimensions (the dimension of $\mathbf{x}$ is $m = 1$ and of $\mathbf{W}$ is $n = 1$): Branin-Hoo and Goldstein-Price; and

---

[2]The code is available at `https://github.com/qphong/BayesOpt-LV`.

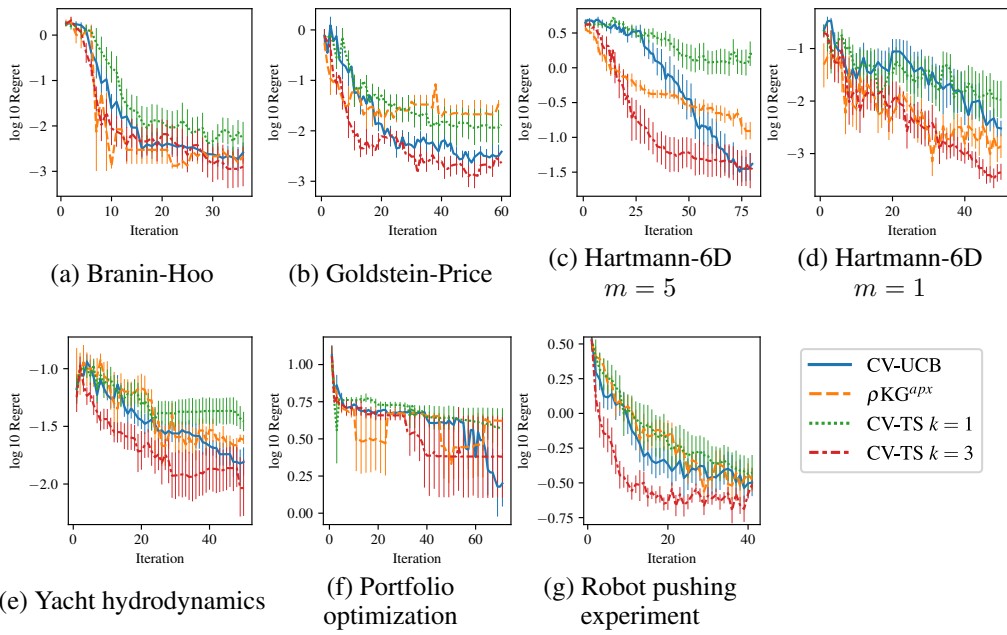

(a) Branin-Hoo     (b) Goldstein-Price     (c) Hartmann-6D $m = 5$     (d) Hartmann-6D $m = 1$

(e) Yacht hydrodynamics     (f) Portfolio optimization     (g) Robot pushing experiment

Figure 2: Plots of the regret against the BO iteration in (a-d) experiments that optimize CVaR of synthetic benchmark functions, (e-g) simulated real-world optimization problems.

with moderate input dimensions ($m = 5$, $n = 1$; and $m = 1$, $n = 5$): Hartmann-6D. The risk level $\alpha$ is set to $0.1$. We observe that CV-UCB achieves comparable performance to the baseline $\rho\text{KG}^{apx}$ in the Branin-Hoo experiment, while it outperforms $\rho\text{KG}^{apx}$ in the Goldstein-Price and Hartmann-6D ($m = 5$) experiments. It is because the evaluation of $\rho\text{KG}^{apx}$ is approximated with samples of $f$ and the nested optimization procedure involved in $\rho\text{KG}^{apx}$ is simplified. CV-TS $k = 1$ does not perform as well as other algorithms do. It is because with few observations, CV-TS $k = 1$ tends towards randomly exploring $\mathbb{X}$ so the maximizer is not accurately located. Furthermore, as the GP hyperparameters are assumed to be unknown in our experiments, they can be incorrectly estimated with few observations, which affects the performance of CV-TS $k = 1$. When we increase the size of the batch query to $3$ in CV-TS $k = 3$, the number of observations increases. In turn, the estimation of the GP hyperparameters and the exploration-exploitation trade-off are improved. Thus, CV-TS $k = 3$ achieves a superior performance. It suggests that CV-TS with a large batch query should be preferred when multiple observations can be obtained, e.g., by running multiple simulations in parallel. In contrast, CV-UCB should be preferred if only one observation is obtained at each BO iteration.

Figs. 2e-g show the results of optimizing CVaR in an optimization problem using the yatch hydrodynamics dataset [5], a portfolio optimization problem [4], and a simulated robot pushing experiment [23]. In the experiment with the yacht hydrodynamics dataset, we would like to minimize the residuary resistance per unit weight of displacement of a yacht by searching for the optimal hull geometry coefficients of the yacht in the face of the uncertainty in the Froude number (the Froude number depends on the real-world environment and we assume that it can be simulated with computers during the optimization). The ground truth function is constructed using the yacht hydrodynamics data set [5]. The dimension of the input variables $\mathbf{x}$ and $\mathbf{W}$ are $m = 5$ and $n = 1$ (the Froude number), respectively. The risk level of CVaR is set to $\alpha = 0.3$. While CV-TS $k = 3$ outperforms the other algorithms significantly thanks to its batch queries, CV-UCB converges to a lower regret than $\rho\text{KG}^{apx}$ does. The portfolio optimization problem is taken from the work of [4] where the evolution of a portfolio over $4$ years is simulated and optimized using open-source market data. The optimization variables (i.e., $\mathbf{x}$) include the risk, the trade aversion, and the holding cost multiplier, while the environmental random variables (i.e., $\mathbf{W}$) include the bid-ask spread and the borrow cost [4]. The risk level of CVaR in this problem is set to $\alpha = 0.2$. We observe that CV-UCB and CV-TS $k = 3$ achieve the best performance by converging to lower regret values than the other algorithms. The simulated robot pushing experiment is taken from [23] whose task is to minimize the distance of a pushed object to a fixed goal location by controlling the robot location and the pushing duration (i.e., $\mathbf{x}$). We adopt the setting of [13] to introduce random perturbations to the robot location (i.e.,

**W**). The risk level of CVaR is set to $\alpha = 0.1$. While all algorithms converge to roughly the same value of the regret in this experiment, we observe that CV-TS $k = 3$ shows its advantage in acquiring more observations by converging faster than the others. Besides, CV-UCB also converges faster than $\rho\text{KG}^{apx}$ in this experiment.

## 6    Conclusion

We propose two BO algorithms to optimize CVaR of a black-box function with theoretical performance guarantee: CV-UCB and CV-TS by taking advantage of a connection between CVaR and VaR and establishing a link between the tail expectation of (non-Gaussian and unbounded support) CVaR and that of the Gaussian-distributed function evaluation. The competitive empirical performance of CV-UCB and CV-TS with batch queries are shown in optimizing CVaR of synthetic benchmark functions and simulated real-world optimization problems. These results recommend the use of CV-UCB to optimize CVaR of a black-box function when only an observation is obtained at each BO iteration and the use of CV-TS with large batch queries when multiple observations can be obtained at each BO iteration.

## Acknowledgments and Disclosure of Funding

This research is supported by A*STAR under its RIE2020 Advanced Manufacturing and Engineering (AME) Programmatic Funds (Award A20H6b0151).

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
