## A    Instantaneous Regret Bound

Conditioned on the event that (8) in Lemma 1 holds (with probability $\geq 1 - \delta$), it follows that

$$c_f(\mathbf{x}_*; \alpha) \leq c_{u_{t-1}}(\mathbf{x}_*; \alpha)$$
$$c_f(\mathbf{x}_t; \alpha) \geq c_{l_{t-1}}(\mathbf{x}_t; \alpha) .$$

Therefore, with probability $\geq 1 - \delta$,

$$
\begin{aligned}
r_t &\triangleq c_f(\mathbf{x}_*; \alpha) - c_f(\mathbf{x}_t; \alpha) \\
&\leq c_{u_{t-1}}(\mathbf{x}_*; \alpha) - c_{l_{t-1}}(\mathbf{x}_t; \alpha) \\
&\leq c_{u_{t-1}}(\mathbf{x}_t; \alpha) - c_{l_{t-1}}(\mathbf{x}_t; \alpha)
\end{aligned}
\tag{18}
$$

where the last inequality is because $\mathbf{x}_t \in \mathrm{argmax}_{\mathbf{x}} \, c_{u_{t-1}}(\mathbf{x}; \alpha)$.

Based on the relationship between CVaR and VaR in (3),

$$
\begin{aligned}
c_{u_{t-1}}(\mathbf{x}_t; \alpha) - c_{l_{t-1}}(\mathbf{x}_t; \alpha) &= \frac{1}{\alpha} \int_0^\alpha v_{u_{t-1}}(\mathbf{x}_t; \alpha') - v_{l_{t-1}}(\mathbf{x}_t; \alpha') \, \mathrm{d}\alpha' \\
&\leq \frac{1}{\alpha} \int_0^\alpha v_{u_{t-1}}(\mathbf{x}_t; \alpha_t) - v_{l_{t-1}}(\mathbf{x}_t; \alpha_t) \, \mathrm{d}\alpha' \\
&= v_{u_{t-1}}(\mathbf{x}_t; \alpha_t) - v_{l_{t-1}}(\mathbf{x}_t; \alpha_t)
\end{aligned}
\tag{19}
$$

where $\alpha_t \in \mathrm{argmax}_{\alpha' \in (0, \alpha]} \, v_{u_{t-1}}(\mathbf{x}_t; \alpha') - v_{l_{t-1}}(\mathbf{x}_t; \alpha')$ (7).

As $\mathbf{w}_t$ is selected as an LV w.r.t. $\alpha_t$, $\mathbf{x}_t$, $l_{t-1}$, and $u_{t-1}$,

$$l_{t-1}(\mathbf{x}_t, \mathbf{w}_t) \leq v_{l_{t-1}}(\mathbf{x}_t; \alpha_t) \leq v_{u_{t-1}}(\mathbf{x}_t; \alpha_t) \leq u_{t-1}(\mathbf{x}_t, \mathbf{w}_t) .$$

Therefore,

$$
\begin{aligned}
v_{u_{t-1}}(\mathbf{x}_t; \alpha_t) - v_{l_{t-1}}(\mathbf{x}_t; \alpha_t) &\leq u_{t-1}(\mathbf{x}_t, \mathbf{w}_t) - l_{t-1}(\mathbf{x}_t, \mathbf{w}_t) \\
&= 2\beta_t^{1/2} \sigma_{t-1}(\mathbf{x}_t, \mathbf{w}_t)
\end{aligned}
\tag{20}
$$

where the last equality is due to (5).

From (18), (19), and (20), we obtain (9), (10), and (11), respectively.

## B    Proof of Theorem 1

From (11) and the nondecreasing property of $\beta_t$, with probability $\geq 1 - \delta$,

$$
\begin{aligned}
R_T = \sum_{t=1}^T r_t &\leq \sum_{t=1}^T 2\beta_t^{1/2} \sigma_{t-1}(\mathbf{x}_t, \mathbf{w}_t) \\
&\leq 2\beta_T^{1/2} \sum_{t=1}^T \sigma_{t-1}(\mathbf{x}_t, \mathbf{w}_t) \\
&\leq 2\beta_T^{1/2} \sqrt{T \sum_{t=1}^T \sigma_{t-1}^2(\mathbf{x}_t, \mathbf{w}_t)}
\end{aligned}
$$

where the last inequality is due to the Cauchy-Schwarz inequality. Assuming $\kappa(\mathbf{x}, \mathbf{w}) \leq 1$ for all $\mathbf{x} \in \mathbb{X}$ and $\mathbf{w} \in \mathbb{W}$, Lemma 5.3 and Lemma 5.4 in [21] show that

$$2\beta_T^{1/2} \sqrt{T \sum_{t=1}^T \sigma_{t-1}^2(\mathbf{x}_t, \mathbf{w}_t)} \leq \sqrt{C_1 T \beta_T \gamma_T} \tag{21}$$

where $C_1 = 8 / \log(1 + \sigma_n^{-2})$ and $\gamma_T$ is the maximum information gain about $f$ that can be obtained by observing any set of $T$ observations. Therefore,

$$R_T \leq \sqrt{C_1 T \beta_T \gamma_T}$$

holds with probability $\geq 1 - \delta$.

## C  Decomposition of $r_t^{\text{Bayes}}$

By selecting $\mathbf{x}_t$ as a sample from the posterior belief of $\mathbf{x}_*$ given $\mathbf{y}_{\mathbf{D}_{t-1}}$, it is noted that the distribution of $\mathbf{x}_t$ and $\mathbf{x}_*$ are the same, i.e., $p(\mathbf{x}_t|\mathbf{y}_{\mathbf{D}_{t-1}}) = p(\mathbf{x}_*|\mathbf{y}_{\mathbf{D}_{t-1}})$. Furthermore, given $\mathbf{y}_{\mathbf{D}_{t-1}}$, $u_{t-1}$ is a deterministic function, so $p(c_{u_{t-1}}(\mathbf{x}_*;\alpha)|\mathbf{y}_{\mathbf{D}_{t-1}}) = p(c_{u_{t-1}}(\mathbf{x}_t;\alpha)|\mathbf{y}_{\mathbf{D}_{t-1}})$ and

$$\mathbb{E}[c_{u_{t-1}}(\mathbf{x}_*;\alpha)] = \mathbb{E}[c_{u_{t-1}}(\mathbf{x}_t;\alpha)] \tag{22}$$

Therefore, following [18], we can decompose $r_t^{\text{Bayes}}$ as follows:

$$\begin{aligned}
r_t^{\text{Bayes}} &\triangleq \mathbb{E}[c_f(\mathbf{x}_*;\alpha) - c_f(\mathbf{x}_t;\alpha)] \\
&= \mathbb{E}[c_f(\mathbf{x}_*;\alpha)] - \mathbb{E}[c_{u_{t-1}}(\mathbf{x}_*;\alpha)] + \mathbb{E}[c_{u_{t-1}}(\mathbf{x}_t;\alpha)] - \mathbb{E}[c_f(\mathbf{x}_t;\alpha)] \quad \text{from (22)} \\
&= \mathbb{E}[c_f(\mathbf{x}_*;\alpha)] - \mathbb{E}[c_{u_{t-1}}(\mathbf{x}_*;\alpha)] + \mathbb{E}[c_{u_{t-1}}(\mathbf{x}_t;\alpha)] - \mathbb{E}[c_{l_{t-1}}(\mathbf{x}_t;\alpha)] \\
&\quad + \mathbb{E}[c_{l_{t-1}}(\mathbf{x}_t;\alpha)] - \mathbb{E}[c_f(\mathbf{x}_t;\alpha)] \\
&= \mathbb{E}[c_{l_{t-1}}(\mathbf{x}_t;\alpha) - c_f(\mathbf{x}_t;\alpha)] + \mathbb{E}[c_f(\mathbf{x}_*;\alpha) - c_{u_{t-1}}(\mathbf{x}_*;\alpha)] \\
&\quad + \mathbb{E}[c_{u_{t-1}}(\mathbf{x}_t;\alpha) - c_{l_{t-1}}(\mathbf{x}_t;\alpha)]
\end{aligned}$$

Since $\mathbb{E}[Z] \leq \mathbb{E}[\max(0, Z)]$ for a random variable $Z$, it follows that

$$\begin{aligned}
r_t^{\text{Bayes}} &\leq \mathbb{E}[\max\big(0, c_{l_{t-1}}(\mathbf{x}_t;\alpha) - c_f(\mathbf{x}_t;\alpha)\big)] + \mathbb{E}[\max\big(0, c_f(\mathbf{x}_*;\alpha) - c_{u_{t-1}}(\mathbf{x}_*;\alpha)\big)] \\
&\quad + \mathbb{E}[c_{u_{t-1}}(\mathbf{x}_t;\alpha) - c_{l_{t-1}}(\mathbf{x}_t;\alpha)] \\
&= \mathbb{E}[\Delta_c^{\text{lower}}(\mathbf{x}_t;\alpha)] + \mathbb{E}[\Delta_c^{\text{upper}}(\mathbf{x}_*;\alpha)] + \mathbb{E}[c_{u_{t-1}}(\mathbf{x}_t;\alpha) - c_{l_{t-1}}(\mathbf{x}_t;\alpha)]
\end{aligned}$$

where $\Delta_c^{\text{lower}}(\mathbf{x}_t;\alpha) \triangleq \max(0, c_{l_{t-1}}(\mathbf{x}_t;\alpha) - c_f(\mathbf{x}_t;\alpha))$ and $\Delta_c^{\text{upper}}(\mathbf{x}_*;\alpha) \triangleq \max(0, c_f(\mathbf{x}_*;\alpha) - c_{u_{t-1}}(\mathbf{x}_*;\alpha))$.

## D  Proof of Lemma 2

$$\begin{aligned}
\mathbb{E}\big[\Delta_c^{\text{lower}}(\mathbf{x};\alpha)\big] &= \mathbb{E}\big[\max(0, c_{l_{t-1}}(\mathbf{x};\alpha) - c_f(\mathbf{x};\alpha))\big] \\
&= \mathbb{E}\left[\max\left(0, \frac{1}{\alpha}\int_0^\alpha \big(v_{l_{t-1}}(\mathbf{x};\alpha') - v_f(\mathbf{x};\alpha')\big)\,\mathrm{d}\alpha'\right)\right] \\
&\leq \mathbb{E}\left[\frac{1}{\alpha}\int_0^\alpha \max\big(0, v_{l_{t-1}}(\mathbf{x};\alpha') - v_f(\mathbf{x};\alpha')\big)\,\mathrm{d}\alpha'\right] \\
&= \frac{1}{\alpha}\int_0^\alpha \mathbb{E}\big[\max\big(0, v_{l_{t-1}}(\mathbf{x};\alpha') - v_f(\mathbf{x};\alpha')\big)\big]\,\mathrm{d}\alpha' \\
&= \frac{1}{\alpha}\int_0^\alpha \mathbb{E}\big[\Delta_v^{\text{lower}}(\mathbf{x};\alpha')\big]\,\mathrm{d}\alpha'
\end{aligned}$$

where $\Delta_v^{\text{lower}}(\mathbf{x};\alpha') \triangleq \max\big(0, v_{l_{t-1}}(\mathbf{x};\alpha') - v_f(\mathbf{x};\alpha')\big)$.

## E  Proof of Lemma 3

We prove the following Lemma 4 which is then used to prove Lemma 5. Lemma 3 follows from Lemma 5.

**Lemma 4.** Let $\mathbb{W}_{l_{t-1}}^{\text{upper}} \triangleq \{\mathbf{w} \in \mathbb{W} : l_{t-1}(\mathbf{x}, \mathbf{w}) \geq v_{l_{t-1}}(\mathbf{x};\alpha')\}$, then $P(\mathbf{W} \in \mathbb{W}_{l_{t-1}}^{\text{upper}}) > 1 - \alpha'$.

*Proof.* By contradiction, if $P(\mathbf{W} \in \mathbb{W}_{l_{t-1}}^{\text{upper}}) \leq 1 - \alpha'$, then

$$P(\mathbf{W} \in \mathbb{W} \setminus \mathbb{W}_{l_{t-1}}^{\text{upper}}) = 1 - P(\mathbf{W} \in \mathbb{W}_{l_{t-1}}^{\text{upper}}) \geq \alpha'.$$

Furthermore, we assume that $|\mathbb{W}|$ is finite, so the above implies that

$$P\left(l_{t-1}(\mathbf{x}, \mathbf{W}) \leq \max_{\mathbf{w} \in \mathbb{W} \setminus \mathbb{W}_{l_{t-1}}^{\text{upper}}} l_{t-1}(\mathbf{x}, \mathbf{w})\right) \geq \alpha'.$$

Therefore, from the definition of VaR,

$$\max_{\mathbf{w} \in \mathbb{W} \setminus \mathbb{W}^{\text{upper}}_{l_{t-1}}} l_{t-1}(\mathbf{x}, \mathbf{w}) \geq v_{l_{t-1}}(\mathbf{x}; \alpha') \,.$$

From the definition of $\mathbb{W}^{\text{upper}}_{l_{t-1}}$, the above implies that

$$\max_{\mathbf{w} \in \mathbb{W} \setminus \mathbb{W}^{\text{upper}}_{l_{t-1}}} l_{t-1}(\mathbf{x}, \mathbf{w}) \in \mathbb{W}^{\text{upper}}_{l_{t-1}} \,.$$

However, $$\max_{\mathbf{w} \in \mathbb{W} \setminus \mathbb{W}^{\text{upper}}_{l_{t-1}}} l_{t-1}(\mathbf{x}, \mathbf{w}) \in \mathbb{W} \setminus \mathbb{W}^{\text{upper}}_{l_{t-1}} \,.$$

Thus, $$\mathbb{W}^{\text{upper}}_{l_{t-1}} \cap \left( \mathbb{W} \setminus \mathbb{W}^{\text{upper}}_{l_{t-1}} \right) \neq \emptyset$$

which is a contradiction. $\qquad\square$

**Lemma 5.** Consider a realization $f_1$ of the black-box function $f$ following the GP posterior belief given $\mathbf{y}_{\mathbf{D}_{t-1}}$ that satisfies

$$v_{l_{t-1}}(\mathbf{x}; \alpha') - v_{f_1}(\mathbf{x}; \alpha') > \omega \tag{23}$$

for $\alpha' \in (0, 1)$, $\mathbf{x} \in \mathbb{X}$, and $\omega \geq 0$. Let $\mathbb{W}^{\text{upper}}_{l_{t-1}} \triangleq \{\mathbf{w} \in \mathbb{W} : l_{t-1}(\mathbf{x}, \mathbf{w}) \geq v_{l_{t-1}}(\mathbf{x}; \alpha')\}$. Then,

$$\exists \mathbf{w}_0 \in \mathbb{W}^{\text{upper}}_{l_{t-1}}, \ v_{l_{t-1}}(\mathbf{x}; \alpha') - f_1(\mathbf{x}, \mathbf{w}_0) > \omega \,.$$

*Proof.* By contradiction, if $\forall \mathbf{w}_0 \in \mathbb{W}^{\text{upper}}_{l_{t-1}}, \ v_{l_{t-1}}(\mathbf{x}; \alpha') - f_1(\mathbf{x}, \mathbf{w}_0) \leq \omega$, i.e., $\forall \mathbf{w}_0 \in \mathbb{W}^{\text{upper}}_{l_{t-1}}, \ f_1(\mathbf{x}, \mathbf{w}_0) + \omega \geq v_{l_{t-1}}(\mathbf{x}; \alpha')$. Furthermore, from (23), $v_{l_{t-1}}(\mathbf{x}; \alpha') > v_{f_1}(\mathbf{x}; \alpha') + \omega$. Therefore,

$$\forall \mathbf{w}_0 \in \mathbb{W}^{\text{upper}}_{l_{t-1}}, \ f_1(\mathbf{x}, \mathbf{w}_0) + \omega > v_{f_1}(\mathbf{x}; \alpha') + \omega \,.$$

Equivalently,

$$\forall \mathbf{w}_0 \in \mathbb{W}^{\text{upper}}_{l_{t-1}}, \ f_1(\mathbf{x}, \mathbf{w}_0) > v_{f_1}(\mathbf{x}; \alpha') \,. \tag{24}$$

By Lemma 4, we have

$$P \left( f_1(\mathbf{x}, \mathbf{W}) \geq \min_{\mathbf{w} \in \mathbb{W}^{\text{upper}}_{l_{t-1}}} f_1(\mathbf{x}, \mathbf{w}) \right) = P(\mathbf{W} \in \mathbb{W}^{\text{upper}}_{l_{t-1}}) > 1 - \alpha' \,. \tag{25}$$

Therefore,

$$1 = P(\mathbf{W} \in \mathbb{W})$$

$$\geq P\left( f_1(\mathbf{x}, \mathbf{W}) \leq v_{f_1}(\mathbf{x}; \alpha') \right) + P \left( f_1(\mathbf{x}, \mathbf{W}) \geq \min_{\mathbf{w} \in \mathbb{W}^{\text{upper}}_{l_{t-1}}} f_1(\mathbf{x}, \mathbf{w}) \right) \text{ due to (24)}$$

$$> \alpha' + 1 - \alpha' \text{ due to (25) and the definition of VaR}$$

$$= 1$$

which is a contradiction. $\qquad\square$

Recall that $\mathbb{W}^{\text{upper}}_{l_{t-1}} \triangleq \{\mathbf{w} \in \mathbb{W} : l_{t-1}(\mathbf{x}, \mathbf{w}) \geq v_{l_{t-1}}(\mathbf{x}; \alpha')\}$. Therefore, $\forall \mathbf{w}_0 \in \mathbb{W}^{\text{upper}}_{l_{t-1}}, l_{t-1}(\mathbf{x}; \alpha') - f_1(\mathbf{x}, \mathbf{w}_0) \geq v_{l_{t-1}}(\mathbf{x}; \alpha') - f_1(\mathbf{x}, \mathbf{w}_0)$. Thus, Lemma 5 implies Lemma 3.

## F  Proof of Theorem 2

Recall $f$ is considered as a random variable, Lemma 3 implies that

$$P(v_{l_{t-1}}(\mathbf{x}; \alpha') - v_f(\mathbf{x}; \alpha') > \omega) \leq P(\exists \mathbf{w} \in \mathbb{W}^{\text{upper}}_{l_{t-1}}, \ l_{t-1}(\mathbf{x}, \mathbf{w}) - f(\mathbf{x}, \mathbf{w}) > \omega)$$

$$\leq \sum_{\mathbf{w} \in \mathbb{W}^{\text{upper}}_{l_{t-1}}} P(l_{t-1}(\mathbf{x}, \mathbf{w}) - f(\mathbf{x}, \mathbf{w}) > \omega) \,. \tag{26}$$

From (16),

$$\mathbb{E}\left[\Delta_v^{\text{lower}}(\mathbf{x};\alpha')\right] = \int_0^\infty P(v_{l_{t-1}}(\mathbf{x};\alpha') - v_f(\mathbf{x};\alpha') > \omega)\,\mathrm{d}\omega$$

$$\leq \int_0^\infty \sum_{\mathbf{w}\in\mathbb{W}_{l_{t-1}}^{\text{upper}}} P(l_{t-1}(\mathbf{x},\mathbf{w}) - f(\mathbf{x},\mathbf{w}) > \omega)\,\mathrm{d}\omega \quad \text{from (26)}$$

$$\leq \sum_{\mathbf{w}\in\mathbb{W}_{l_{t-1}}^{\text{upper}}} \int_0^\infty P(l_{t-1}(\mathbf{x},\mathbf{w}) - f(\mathbf{x},\mathbf{w}) > \omega)\,\mathrm{d}\omega$$

$$= \sum_{\mathbf{w}\in\mathbb{W}_{l_{t-1}}^{\text{upper}}} \mathbb{E}\left[\max(0, l_{t-1}(\mathbf{x},\mathbf{w}) - f(\mathbf{x},\mathbf{w}))\right] . \tag{27}$$

Since $l_{t-1}(\mathbf{x},\mathbf{w}) - f(\mathbf{x},\mathbf{w})$ is a Gaussian random variable with mean $l_{t-1}(\mathbf{x},\mathbf{w}) - \mu_{t-1}(\mathbf{x},\mathbf{w}) = -\beta_t^{1/2}\sigma_{t-1}(\mathbf{x},\mathbf{w})$ and variance $\sigma_{t-1}^2(\mathbf{x},\mathbf{w})$, it follows that

$$\mathbb{E}\left[\max(0, l_{t-1}(\mathbf{x},\mathbf{w}) - f(\mathbf{x},\mathbf{w}))\right]$$

$$= \int_0^\infty \frac{\omega}{\sigma_{t-1}(\mathbf{x},\mathbf{w})\sqrt{2\pi}} \exp\left(-\frac{(\omega + \beta_t^{1/2}\sigma_{t-1}(\mathbf{x},\mathbf{w}))^2}{2\sigma_{t-1}^2(\mathbf{x},\mathbf{w})}\right)\,\mathrm{d}\omega$$

$$\leq \frac{\sigma_{t-1}(\mathbf{x},\mathbf{w})}{\sqrt{2\pi}} \exp\left(\frac{-\beta_t}{2}\right)$$

$$= \frac{\sigma_{t-1}(\mathbf{x},\mathbf{w})}{\sqrt{2\pi}} \frac{\delta}{|\mathbb{X}||\mathbb{W}|\pi_t} \quad \text{since } \beta_t = 2\log(|\mathbb{X}||\mathbb{W}|\pi_t/\delta) \text{ in Lemma 1}$$

$$\leq \frac{\delta}{|\mathbb{X}||\mathbb{W}|\sqrt{2\pi}}\pi_t^{-1} \tag{28}$$

where the last inequality is due to the assumption $\kappa(\mathbf{x},\mathbf{w}) \leq 1\ \forall(\mathbf{x},\mathbf{w}) \in \mathbb{X}\times\mathbb{W}$.

From (27) and (28),

$$\mathbb{E}\left[\Delta_v^{\text{lower}}(\mathbf{x};\alpha')\right] \leq |\mathbb{W}_{l_{t-1}}^{\text{upper}}|\frac{\delta}{|\mathbb{X}||\mathbb{W}|\sqrt{2\pi}}\pi_t^{-1} \leq \frac{\delta}{|\mathbb{X}|\sqrt{2\pi}}\pi_t^{-1} . \tag{29}$$

Similar to the bound of $\Delta_v^{\text{lower}}(\mathbf{x};\alpha')$, we can bound $\Delta_v^{\text{upper}}(\mathbf{x};\alpha')$ by considering the set $\mathbb{W}_{u_{t-1}}^{\text{lower}} \triangleq \{\mathbf{w}\in\mathbb{W} : u_{t-1}(\mathbf{x},\mathbf{w}) \leq v_{u_{t-1}}(\mathbf{x};\alpha')\}$:

$$\mathbb{E}\left[\Delta_v^{\text{upper}}(\mathbf{x};\alpha')\right] \leq |\mathbb{W}_{u_{t-1}}^{\text{lower}}|\frac{\delta}{|\mathbb{X}||\mathbb{W}|\sqrt{2\pi}}\pi_t^{-1} \leq \frac{\delta}{|\mathbb{X}|\sqrt{2\pi}}\pi_t^{-1} . \tag{30}$$

From (15), (29) and (30), we have

$$\mathbb{E}[\Delta_c^{\text{lower}}(\mathbf{x};\alpha)] \leq \frac{\delta}{|\mathbb{X}|\sqrt{2\pi}}\pi_t^{-1} \tag{31}$$

$$\mathbb{E}[\Delta_c^{\text{upper}}(\mathbf{x};\alpha)] \leq \frac{\delta}{|\mathbb{X}|\sqrt{2\pi}}\pi_t^{-1} . \tag{32}$$

From (13), (14), (31), and (32), $r_t^{\text{Bayes}}$ can be bounded:

$$r_t^{\text{Bayes}} \leq \frac{\delta\sqrt{2}}{|\mathbb{X}|\sqrt{\pi}}\pi_t^{-1} + 2\beta_t^{1/2}\sigma_{t-1}(\mathbf{x}_t,\mathbf{w}_t) \tag{33}$$

**Algorithm 2** CV-TS with batch queries for optimizing CVaR of a black-box function

1: **Input:** $k$, $\mathbb{X}$, $\mathbb{W}$, initial observation $\mathbf{y}_{\mathbf{D}_0}$, prior $\mu_0 = 0, \sigma_n, \kappa$
2: **for** $t = 1, 2, \ldots$ **do**
3:     Sample $k$ functions $(f_j)_{j=1}^k$ from the GP posterior belief given $\mathbf{y}_{\mathbf{D}_{k(t-1)}}$
4:     **for** $j = 1, 2, \ldots, k$ **do**
5:         Select $\mathbf{x}_{k(t-1)+j} \in \arg\max_{\mathbf{x}} c_{f_j}(\mathbf{x}; \alpha)$
6:         Find $\alpha_{k(t-1)+j} \in \arg\max_{\alpha' \in (0, \alpha]} v_{u_{k(t-1)}}(\mathbf{x}_t; \alpha') - v_{l_{k(t-1)}}(\mathbf{x}_t; \alpha')$
7:         Given $\alpha_{k(t-1)+j}$, select $\mathbf{w}_{k(t-1)+j}$ as an LV w.r.t. $\mathbf{x}_{k(t-1)+j}$, $u_{k(t-1)}$, and $l_{k(t-1)}$.
8:     **end for**
9:     Incorporate new observations at the batch query: $\mathbf{y}_{\mathbf{D}_{kt}} = \mathbf{y}_{\mathbf{D}_{k(t-1)}} \cup \{y(\mathbf{x}_i, \mathbf{w}_i)\}_{i=k(t-1)+1}^{kt}$
10:     Update the GP posterior belief given $\mathbf{y}_{\mathbf{D}_{kt}}$ to obtain $\mu_{kt}$ and $\sigma_{kt}^2$
11: **end for**

---

Therefore, the Bayesian cumulative regret is bounded by:

$$
\begin{aligned}
R_t^{\text{Bayes}} &= \mathbb{E}\left[\sum_{t=1}^T r_t^{\text{Bayes}}\right] \\
&\leq \mathbb{E}\left[\frac{\delta\sqrt{2}}{|\mathbb{X}|\sqrt{\pi}} \sum_{t=1}^T \pi_t^{-1} + \sum_{t=1}^T 2\beta_t^{1/2} \sigma_{t-1}(\mathbf{x}_t, \mathbf{w}_t)\right] \\
&\leq \frac{\delta\sqrt{2}}{|\mathbb{X}|\sqrt{\pi}} + \sqrt{C_1 T \beta_T \gamma_T}
\end{aligned}
\tag{34}
$$

where the last inequality is because $\sum_{t=1}^T \pi_t^{-1} \leq \sum_{t \geq 1} \pi_t^{-1} = 1$ (in Lemma 1) and $\sum_{t=1}^T 2\beta_t^{1/2} \sigma_{t-1}(\mathbf{x}_t, \mathbf{w}_t) \leq \sqrt{C_1 T \beta_T \gamma_T}$ shown in Appendix B.

## G    CV-TS with Batch Queries

Let us consider CV-TS with a batch query of size $k$ at each iteration. To simplify the notation, let us assume that the set of initial observations is empty, i.e., $\mathbf{D}_0 = \emptyset$. Following the indexing of observed inputs from [11], inputs in the first batch query (at BO iteration $t = 1$) are indexed by $i = 1, \ldots, k$, inputs in the second batch query (at BO iteration $t = 2$) are indexed by $i = k + 1, \ldots, 2k$, and so on. We denote $\mathbf{D}_i \triangleq \{\mathbf{x}_j\}_{j=1}^i$. Then, the set of observed inputs at index $i$ is $\mathbf{D}_{k\lfloor \frac{i-1}{k} \rfloor}$ where $\lfloor \frac{i-1}{k} \rfloor$ is the greatest integer less than or equal to $\frac{i-1}{k}$. At BO iteration $t$, CV-TS selects a batch query $\{\mathbf{x}_i\}_{i=k(t-1)+1}^{kt}$ by drawing $k$ samples of the maximizer of $c_f(\mathbf{x}; \alpha)$ given observations at $\{\mathbf{x}_j\}_{j=1}^{k(t-1)}$ (i.e., $\mathbf{D}_{k(t-1)}$).

Since at index $i$ we only have access to observations $\mathbf{y}_{\mathbf{D}_{k\lfloor \frac{i-1}{k} \rfloor}}$, the confidence bound of $f(\mathbf{x}, \mathbf{w})$ at index $i$ is

$$
\left[l_{k\lfloor \frac{i-1}{k} \rfloor}(\mathbf{x}, \mathbf{w}), u_{k\lfloor \frac{i-1}{k} \rfloor}(\mathbf{x}, \mathbf{w})\right] .
$$

At index $i$, given $\mathbf{D}_{k\lfloor \frac{i-1}{k} \rfloor}$, the distribution of $\mathbf{x}_i$ is the same that that of $\mathbf{x}_*$ (due to the selection strategy of CV-TS), so $\mathbb{E}\left[c_{u_{k\lfloor \frac{i-1}{k} \rfloor}}(\mathbf{x}_*; \alpha)\right] = \mathbb{E}\left[c_{u_{k\lfloor \frac{i-1}{k} \rfloor}}(\mathbf{x}_i; \alpha)\right]$. Let us use $T$ to denote the total number of observations. Then, $T$ is a multiple of $k$ because there are $k$ observations at each BO iteration and we assume that $|\mathbf{D}_0| = \emptyset$. We can decompose the Bayesian cumulative regret of CV-TS

with a batch query of size $k$, denoted as $R_T^{\text{Bayes}}(k)$:

$$R_T^{\text{Bayes}}(k) = \mathbb{E}\left[\sum_{i=1}^{T} c_f(\mathbf{x}_*;\alpha) - c_f(\mathbf{x}_i;\alpha)\right]$$

$$= \mathbb{E}\left[\sum_{i=1}^{T} \underbrace{\mathbb{E}\left[c_f(\mathbf{x}_*;\alpha) - c_{u_{k\lfloor\frac{i-1}{k}\rfloor}}(\mathbf{x}_*;\alpha)|\mathbf{y}_{\mathbf{D}_{k\lfloor\frac{i-1}{k}\rfloor}}\right]}_{A_0}\right]$$

$$+ \mathbb{E}\left[\underbrace{\sum_{i=1}^{T} c_{u_{k\lfloor\frac{i-1}{k}\rfloor}}(\mathbf{x}_i;\alpha) - c_{l_{k\lfloor\frac{i-1}{k}\rfloor}}(\mathbf{x}_i;\alpha)}_{B}\right]$$

$$+ \mathbb{E}\left[\sum_{i=1}^{T} \underbrace{\mathbb{E}\left[c_{l_{k\lfloor\frac{i-1}{k}\rfloor}}(\mathbf{x}_i;\alpha) - c_f(\mathbf{x}_i;\alpha)|\mathbf{y}_{\mathbf{D}_{k\lfloor\frac{i-1}{k}\rfloor}}\right]}_{A_1}\right].$$

Similar to (31) and (32), $A_0$ and $A_1$ can be bounded by the tail expectations of CVaR which are bounded by $\frac{\delta}{|\mathbb{X}|\sqrt{2\pi}}\pi_{k\lfloor\frac{i-1}{k}\rfloor}^{-1}$. Then,

$$\mathbb{E}\left[\sum_{i=1}^{T} A_0\right] \leq \mathbb{E}\left[\sum_{i=1}^{T} \frac{\delta}{|\mathbb{X}|\sqrt{2\pi}}\pi_{k\lfloor\frac{i-1}{k}\rfloor}^{-1}\right] = \frac{\delta}{|\mathbb{X}|\sqrt{2\pi}}\sum_{i=1}^{T}\pi_{k\lfloor\frac{i-1}{k}\rfloor}^{-1} \leq \frac{\delta}{|\mathbb{X}|\sqrt{2\pi}}\sum_{t\geq 1}\pi_t^{-1} = \frac{\delta}{|\mathbb{X}|\sqrt{2\pi}} \tag{35}$$

$$\mathbb{E}\left[\sum_{i=1}^{T} A_1\right] \leq \mathbb{E}\left[\sum_{i=1}^{T} \frac{\delta}{|\mathbb{X}|\sqrt{2\pi}}\pi_{k\lfloor\frac{i-1}{k}\rfloor}^{-1}\right] = \frac{\delta}{|\mathbb{X}|\sqrt{2\pi}}\sum_{i=1}^{T}\pi_{k\lfloor\frac{i-1}{k}\rfloor}^{-1} \leq \frac{\delta}{|\mathbb{X}|\sqrt{2\pi}}\sum_{t\geq 1}\pi_t^{-1} \leq \frac{\delta}{|\mathbb{X}|\sqrt{2\pi}}. \tag{36}$$

The term $B$ is bounded as follows.

$$B = \mathbb{E}\left[\sum_{i=1}^{T} c_{u_{k\lfloor\frac{i-1}{k}\rfloor}}(\mathbf{x}_i;\alpha) - c_{l_{k\lfloor\frac{i-1}{k}\rfloor}}(\mathbf{x}_i;\alpha)\right]$$

$$\leq \mathbb{E}\left[\sum_{i=1}^{T} 2\beta_{k\lfloor\frac{i-1}{k}\rfloor+1}^{1/2}\sigma_{k\lfloor\frac{i-1}{k}\rfloor}(\mathbf{x}_i,\mathbf{w}_i)\right] \tag{37}$$

$$= \mathbb{E}\left[\sum_{i=1}^{k} 2\beta_{k\lfloor\frac{i-1}{k}\rfloor+1}^{1/2}\sigma_{k\lfloor\frac{i-1}{k}\rfloor}(\mathbf{x}_i,\mathbf{w}_i) + \sum_{i=k+1}^{T} 2\beta_{k\lfloor\frac{i-1}{k}\rfloor+1}^{1/2}\sigma_{k\lfloor\frac{i-1}{k}\rfloor}(\mathbf{x}_i,\mathbf{w}_i)\right]$$

$$\leq \mathbb{E}\left[\sum_{i=1}^{k} 2\beta_1^{1/2} + \sum_{i=k+1}^{T} 2\beta_{k\lfloor\frac{T-1}{k}\rfloor+1}^{1/2}\sigma_{i-k-1}(\mathbf{x}_i,\mathbf{w}_i)\right] \tag{38}$$

$$\leq 2k\beta_1^{1/2} + \mathbb{E}\left[2\beta_{k\lfloor\frac{T-1}{k}\rfloor+1}^{1/2}\sum_{i=1}^{T-k}\sigma_{i-1}(\mathbf{x}_i,\mathbf{w}_i)\right]$$

$$\leq 2k\beta_1^{1/2} + \mathbb{E}\left[2\beta_{k\lfloor\frac{T-1}{k}\rfloor+1}^{1/2}\sqrt{(T-k)\sum_{i=1}^{T-k}\sigma_{i-1}^2(\mathbf{x}_i,\mathbf{w}_i)}\right] \tag{39}$$

$$\leq 2k\beta_1^{1/2} + \sqrt{\frac{8(T-k)\beta_{k\lfloor\frac{T-1}{k}\rfloor+1}\gamma_{T-k}}{\log(1+\sigma_n^{-2})}} \tag{40}$$

$$\leq 2k\beta_1^{1/2} + \sqrt{C_1(T-k)\beta_{T-k+1}\gamma_{T-k}} \tag{41}$$

where

- (37) is because of (19) and (20).
- (38) is because $\beta_t$ is nondecreasing, $\kappa(\mathbf{x}, \mathbf{w}) \leq 1$ (our assumption), and $\sigma_{i-k-1} \geq \sigma_{k\lfloor \frac{i-1}{k} \rfloor}$ for $i = k+1, \ldots, T$ (since $\mathbf{D}_{i-k-1} \subset \mathbf{D}_{k\lfloor \frac{i-1}{k} \rfloor}$).
- (39) is because of the Cauchy-Schwarz inequality.
- (40) is because of Lemma 5.3 and Lemma 5.4 in [21] and our assumption $\kappa(\mathbf{x}, \mathbf{w}) \leq 1$.

From (35), (36), and (41), the Bayesian cumulative regret is bounded by:

$$R_T^{\text{Bayes}}(k) \leq \frac{\delta\sqrt{2}}{|\mathbb{X}|\sqrt{\pi}} + 2k\beta_1^{1/2} + \sqrt{C_1(T-k)\beta_{T-k+1}\gamma_{T-k}} \,. \tag{42}$$

Recall the Bayesian cumulative regret bound for CV-TS with single queries (i.e., $k = 1$) in (34):

$$R_T^{\text{Bayes}} \leq \frac{\delta\sqrt{2}}{|\mathbb{X}|\sqrt{\pi}} + \sqrt{C_1 T \beta_T \gamma_T} \,. \tag{43}$$

Hence, the average of the Bayesian cumulative regret for CV-TS with single queries, $R_T^{\text{Bayes}}/T$, and batch queries, $R_T^{\text{Bayes}}(k)/T$, are similar, especially when the number of observations $T$ is large (so that $2k\beta_1^{1/2}/T$ vanishes).

## H   A Thompson Sampling Approach to Optimize VaR of Black-Box Functions

### H.1   Algorithm

We present an algorithm to optimize VaR $v_f(\mathbf{x}; \alpha)$ of a black-box function $f(\mathbf{x}, \mathbf{W})$. Unlike the existing V-UCB algorithm in [13] that is based on the upper confidence bound, this algorithm is based on the Thompson sampling approach which is called V-TS (Algorithm 3).

Following the popular Thompson sampling approach (or posterior sampling [18]), V-TS selects $\mathbf{x}_t$ as a sample of the maximizer of VaR $v_f(\mathbf{x}; \alpha)$ by: (line 4 of Algorithm 3) using the random Fourier feature approximation method [16] to draw a function sample $f_1$ from the GP posterior belief given $\mathbf{y}_{\mathbf{D}_{t-1}}$ and (line 5 of Algorithm 3) assigning the maximizer of $v_{f_1}(\mathbf{x}; \alpha)$ to $\mathbf{x}_t$.

Given the selected $\mathbf{x}_t$, we select $\mathbf{w}_t$ to reduce the uncertainty of VaR $v_f(\mathbf{x}_t; \alpha)$ quantified by the size of its confidence bound $v_{u_{t-1}}(\mathbf{x}_t; \alpha) - v_{l_{t-1}}(\mathbf{x}_t; \alpha)$. Following the same approach in Sec. 3.2, we select $\mathbf{w}_t$ as an LV w.r.t. $\alpha$, $\mathbf{x}_t$, $l_{t-1}$, and $u_{t-1}$ (line 7 of Algorithm 3). If there are multiple LVs, we select the LV with the maximum probability $p(\mathbf{W})$. It is a heuristic to improve the empirical performance suggested by [13].

Like CV-TS with batch queries (Appendix G), V-TS can also be extended to handle a batch query of size $k$, i.e., V-TS selects a batch of $k$ inputs to query for their observations at each BO iteration. This batch of $k$ inputs are obtained by: drawing $k$ samples of the maximizer of $v_f(\mathbf{x}; \alpha)$ given $\mathbf{y}_{\mathbf{D}_{t-1}}$ and finding the corresponding $k$ LVs w.r.t. these $k$ samples, $\alpha$, $l_{t-1}$, and $u_{t-1}$.

### H.2   Theoretical Analysis

Let us consider V-TS that selects a single query at each BO iteration (Algorithm 3). We would like to show that the Bayesian cumulative regret of V-TS is sublinear. Let $\mathbf{x}_* \in \arg\max_{\mathbf{x} \in \mathbb{X}} v_f(\mathbf{x}; \alpha)$. The Bayesian cumulative regret can be expressed as

$$R_T^{\text{Bayes}} = \mathbb{E}\left[\sum_{t=1}^{T} v_f(\mathbf{x}_*; \alpha) - v_f(\mathbf{x}_t; \alpha)\right]$$

$$= \mathbb{E}\left[\sum_{t=1}^{T} \mathbb{E}\left[v_f(\mathbf{x}_*; \alpha) - v_f(\mathbf{x}_t; \alpha)|\mathbf{y}_{\mathbf{D}_{t-1}}\right]\right] \,.$$

**Algorithm 3** V-TS: A BO Algorithm for optimizing VaR of a black-box function
___

1: **Input:** $\mathbb{X}$, $\mathbb{W}$, initial observation $\mathbf{y}_{\mathbf{D}_0}$, prior $\mu_0 = 0, \sigma_n, \kappa$
2: **for** $t = 1, 2, \dots$ **do**
3:     {*Selecting* $\mathbf{x}_t$}
4:     Sample a function $f_1$ from the GP posterior belief given $\mathbf{y}_{\mathbf{D}_{t-1}}$
5:     Select $\mathbf{x}_t \in \underset{\mathbf{x}}{\operatorname{argmax}}\, v_{f_1}(\mathbf{x}; \alpha)$
6:     {*Selecting* $\mathbf{w}_t$}
7:     Select $\mathbf{w}_t$ as an LV w.r.t. $\alpha$, $\mathbf{x}_t$, $u_{t-1}$, and $l_{t-1}$
8:     {*Collecting data and updating GP*}
9:     Incorporate new observation at input query: $\mathbf{y}_{\mathbf{D}_t} = \mathbf{y}_{\mathbf{D}_{t-1}} \cup \{y(\mathbf{x}_t, \mathbf{w}_t)\}$
10:    Update the GP posterior belief given $\mathbf{y}_{\mathbf{D}_t}$
11: **end for**
___

The expectation $\mathbb{E}\left[v_f(\mathbf{x}_*; \alpha) - v_f(\mathbf{x}_t; \alpha) | \mathbf{y}_{\mathbf{D}_{t-1}}\right]$ can be decomposed into (in a similar fashion to (13) where we omit $\mathbf{y}_{\mathbf{D}_{t-1}}$ to ease the notational clutter):

$$\mathbb{E}\left[v_f(\mathbf{x}_*; \alpha) - v_{u_{t-1}}(\mathbf{x}_*; \alpha)\right] + \mathbb{E}\left[v_{u_{t-1}}(\mathbf{x}_*; \alpha) - v_{l_{t-1}}(\mathbf{x}_t; \alpha)\right] + \mathbb{E}\left[v_{l_{t-1}}(\mathbf{x}_t; \alpha) - v_f(\mathbf{x}_t; \alpha)\right]$$

$$= \mathbb{E}\left[v_f(\mathbf{x}_*; \alpha) - v_{u_{t-1}}(\mathbf{x}_*; \alpha)\right] + \mathbb{E}\left[v_{u_{t-1}}(\mathbf{x}_t; \alpha) - v_{l_{t-1}}(\mathbf{x}_t; \alpha)\right] \tag{44}$$

$$\quad + \mathbb{E}\left[v_{l_{t-1}}(\mathbf{x}_t; \alpha) - v_f(\mathbf{x}_t; \alpha)\right]$$

$$\leq \mathbb{E}\left[\max\left(0, v_f(\mathbf{x}_*; \alpha) - v_{u_{t-1}}(\mathbf{x}_*; \alpha)\right)\right] + \mathbb{E}\left[v_{u_{t-1}}(\mathbf{x}_t; \alpha) - v_{l_{t-1}}(\mathbf{x}_t; \alpha)\right]$$

$$\quad + \mathbb{E}\left[\max\left(0, v_{l_{t-1}}(\mathbf{x}_t; \alpha) - v_f(\mathbf{x}_t; \alpha)\right)\right]$$

$$= \mathbb{E}\left[\Delta_v^{\text{upper}}(\mathbf{x}_*; \alpha)\right] + \mathbb{E}\left[v_{u_{t-1}}(\mathbf{x}_t; \alpha) - v_{l_{t-1}}(\mathbf{x}_t; \alpha)\right] + \mathbb{E}\left[\Delta_v^{\text{lower}}(\mathbf{x}_t; \alpha)\right]$$

where (44) is because we select $\mathbf{x}_t$ as a sample of the maximizer of $v_f(\mathbf{x}; \alpha)$ given $\mathbf{y}_{\mathbf{D}_{t-1}}$ (lines 4-5 of Algorithm 3), i.e., the distribution of $\mathbf{x}_t$ is the same as that of $\mathbf{x}_*$ given $\mathbf{y}_{\mathbf{D}_{t-1}}$.

The bounds of $\mathbb{E}\left[\Delta_v^{\text{lower}}(\mathbf{x}_*; \alpha)\right]$ and $\mathbb{E}\left[\Delta_v^{\text{upper}}(\mathbf{x}_t; \alpha)\right]$ are obtained from (29) and (30), while the bound of $\mathbb{E}\left[v_{u_{t-1}}(\mathbf{x}_t; \alpha) - v_{l_{t-1}}(\mathbf{x}_t; \alpha)\right]$ is obtained from (20) (since $\mathbf{w}_t$ is selected as an LV w.r.t. $\alpha$, $\mathbf{x}_t$, $l_{t-1}$, and $u_{t-1}$). Therefore,

$$R_T^{\text{Bayes}} \leq \mathbb{E}\left[\sum_{t=1}^T \frac{\delta\sqrt{2}}{|\mathbb{X}|\sqrt{\pi}} \pi_t^{-1} + 2\beta_t^{1/2}\sigma_{t-1}(\mathbf{x}_t, \mathbf{w}_t)\right] \tag{45}$$

$$\leq \frac{\delta\sqrt{2}}{|\mathbb{X}|\sqrt{\pi}} + \sqrt{C_1 T \beta_T \gamma_T} \tag{46}$$

where $C_1$, $\beta_T$, $\delta$, $\gamma_T$ are elaborated in Theorem 2.

## I   Experimental Details

We use the Matérn $5/2$ kernel,

$$\kappa(\mathbf{x}, \mathbf{w}; \mathbf{x}', \mathbf{w}') = \sigma_s^2\left(1 + \sqrt{5}r + \frac{5r^2}{3}\right)\exp\left(-\sqrt{5}r\right) \tag{47}$$

where $r^2 \triangleq (\mathbf{x} - \mathbf{x}')^\top \mathbf{L}_x^{-2}(\mathbf{x} - \mathbf{x}') + (\mathbf{w} - \mathbf{w}')^\top \mathbf{L}_w^{-2}(\mathbf{w} - \mathbf{w}')$ is the squared scaled Euclidean distance between $[\mathbf{x}, \mathbf{w}]$ and $[\mathbf{x}', \mathbf{w}']$, $\mathbf{L}_x \triangleq \operatorname{diag}[l_1, \dots, l_m]$ and $\mathbf{L}_w \triangleq \operatorname{diag}[l_{m+1}, \dots, l_{m+n}]$ are the length-scales.

At BO iteration $t$, the GP hyperparameters (i.e., $\sigma_s^2$, $\mathbf{L}_x$, and $\mathbf{L}_w$) and the noise variance $\sigma_n^2$ are learned by maximizing the likelihood of the observations $\mathbf{y}_{\mathbf{D}_{t-1}}$. We impose a Gamma prior distribution of shape 1.1 and scale 0.5 over the noise variance and initialize the noise variance $\sigma_n^2$ at the mode of its prior distribution, i.e., 0.05 (which is adopted from the implementation of [4]).

The domains of all input dimensions in the experiments are standardized to the range $[0, 1]$. There are 3 initial observations for the experiments with the Branin-Hoo and Goldstein-Price functions, and 20 initial observations for the experiment with the Hartmann-6D function with $m = 5$ and 10 initial

observations for the experiment with the Hartmann-6D function with $m = 1$. The sizes $|\mathbb{W}|$ in the experiments with Branin-Hoo, Goldstein-Price, Hartmann-6D $m = 5$, and Hartmann-6D $m = 1$ are 30, 50, 15, and 243, respectively. We perform experiments with both uniform distributions of $\mathbf{W}$ (in the experiments with Branin-Hoo and Goldstein-Price) and a non-uniform distribution of $\mathbf{W}$ (in the experiment with Hartmann-6D). The non-uniform distribution is a discretized Gaussian distribution with mean 0.5 and standard deviation 0.2 over the support of $\mathbf{W}$.

In the yacht hydrodynamics experiment, we would like to minimize the residuary resistance per unit weight of displacement of a yacht by searching for the optimal hull geometry coefficients of the yacht in the face of the uncertainty in the Froude number (the Froude number depends on the real-world environment and we assume that it can be simulated with computers during the optimization). The ground truth function is constructed using the yacht hydrodynamics data set [5]. The dimension of the input variables $\mathbf{x}$ and $\mathbf{W}$ are $m = 5$ and $n = 1$ (the Froude number), respectively. The environmental random variable $\mathbf{W}$ follows a discrete uniform random variable over the support of 15 values.

The simulated robot pushing experiment is taken from [23]. The simulation returns the location of a pushed object given the robot's location and the pushing duration, i.e., $\mathbf{x}$. The locations are 2 dimensional and standardized in $[0, 1]^2$. We follow the setting in [13] to perturb the robot's location with $\mathbf{W}$ following a discrete uniform distribution over 64 points in $[0, 1]^2$. The location of the pushed object returned by the simulation is added with a Gaussian noise of variance 0.0001 to generate noisy observations. There are 30 initial observations, i.e., $|\mathbf{D}_0| = 30$.

The portfolio optimization problem is taken from [4]. The objective function is the average daily return over a period of 4 years (obtained by a simulation) given the risk and trade aversion parameters, and the holding cost multiplier. The environmental random variables $\mathbf{W}$ include the bid-ask spread and the borrow cost. The distribution of $\mathbf{W}$ is a discretized Gaussian distribution with mean 0.5 and standard deviation 0.15 over 25 points in $[0.25, 0.75]^2$. The average daily returns are added with a Gaussian noise of variance 0.0001 to generate noisy observations. There are 30 initial observations, i.e., $|\mathbf{D}_0| = 30$.