# OpenReview forum: "Optimizing Conditional Value-At-Risk of Black-Box Functions"
_NeurIPS.cc/2021/Conference — NeurIPS 2021 Poster_

### Official Review · Reviewer_HzFE · 2021-07-05

**Rating:** 6
**Confidence:** 4

**Summary:**

This paper considers a robust/risk-averse Bayesian optimization problem. In particular, the authors focus on maximizing conditional value at risk (CVaR), when the objective is unknown.
They propose two algorithms based on UCB and Thompson sampling strategies. Both algorithms are tested on synthetic datasets/tasks.


**Limitations And Societal Impact:**

Yes.

**Main Review:**

The paper is clearly written and easy to understand. It contributes novel algorithms and tools (mostly, when it comes to the introduced TS strategy) for robust Bayesian optimization (BO). It builds upon previous robust BO works in which the goal is to optimize the unknown function subject to uncontrollable (at test time) covariate. In general, this problem is interesting and is of significant interest in practical applications. Different from the previous works, that study worst-case robust optimization, DRBO, mixed strategies, and VaR objectives, this work considers the CVaR objective.

As a related work, I’d expect more comments on the relation between CVaR formulation and DRO problem (with particular ambiguity sets) and the fact that DRO models have an equivalent risk-averse optimization problem (see, e.g., Rahimian and Mehrotra, DRO review, Section 3).
I’m wondering to what extent can we then utilize the existing DRBO algorithms? And, in such a case, wouldn’t they at least be proper baselines to compare against?

162:166: I think that this paragraph needs to be more precise. Before this paragraph on time complexity, you haven’t specified the set X (later on, X is considered to be finite). N_train and gradient-based optimization of the acquisition function is not explained. If X is not finite, then we are talking non-convex continuous optimization (this is fine, since non-convex acquisition functions are standard in BO, and for the sake of theory we assume that we can optimize them). The sentence, “the number of gradient descent steps to find maximizer” sounds like you’re solving a concave problem.

At first selecting alpha_t in the algorithm given that alpha is fixed was strange to me, e.g., why learn at different levels of risk alpha if the learner knows the true parameter value. By following your regret derivation in the appendix, I could see why this is necessary for the analysis to work. Do you think that there is a way to directly select w_t as a function of alpha and x_t and not alpha_t? E.g., in the previous robust BO works, such as the one cited below or, e.g., [1], the second variable is selected to maximize the uncertainty given the previously selected x_t.

The concept of lacing values is very important and crucial in the analysis. It’d be good perhaps to explain/summarize the proof of existence (perhaps, in the appendix) from [12] to improve the clarity of the paper.

Can you comment on your analysis when X is not finite? For example, this assumption is used in the proof of Theorem 2. Also, having X being finite and having the main results expressed in terms of gamma_T, with the additional comment of gamma_T being sublinear for most of the kernels is a bit informal.

Figure 1 nicely explains the difference between VaR and CVaR. I like the joint pseudocode presentation of TS and UCB with the common selection of w_t. I find the performed experiments to be very standard (as in other BO and robust BO works) and not exciting.

Some minor comments:

-- Abstract: “an interesting connection between CVaR and VaR”; Aren’t we only talking about one of the definitions of CVaR in terms of VaR? Can you clarify the “interesting connection”?

-- It’d be interesting to also learn about the distribution of W simultaneously as was also considered in some of the robust BO papers.

-- Another relevant robust BO work is: “Mixed Strategies for Robust Optimization of Unknown Objectives” from Sessa et al. where instead of selecting a single robust design as in your work, the authors seek to discover a robust distribution.

-- 108: The considered noise model is homoscedastic, so perhaps avoid writing epsilon(x,w).

-- Algorithm 1: alpha is an input.

-- An interesting research question, is can you get similar regret guarantees and adapt in case the true alpha is unknown?

-- Mathematical definition of the maximum information gain is missing.


**Time Spent Reviewing:**

4

---

> ### Author Response · Authors · 2021-08-10
> **Author Response**
>
> Thank you for your valuable feedbacks and the interesting connection between risk-averse optimization formulation and DRO you pointed out in Rahimian and Mehrotra, DRO review, Section 3. To address some of your comments, we would like to provide further information, a toy example, and a real-world experiment to justify the strength of our paper as follows.
>
>
>
> Regarding the connection between CVaR formulation and DRO in Rahimian and Mehrotra, DRO review, Section 3, we observe that while DRO can be framed as a risk-averse optimization problem, the reverse is not always true. Let us consider the DRO problem defined as
>
> $\quad \inf_{\mathbf{x} \in \mathcal{X}} \sup_{P \in \mathcal{P}} \mathbb{E}_P [h(\mathbf{x}, \tilde{\xi})]$.
>
> It can be written as a risk-averse optimization problem:
>
> $\quad \inf_{\mathbf{x} \in \mathcal{X}} \sup_{\mathbf{p} \in \mathcal{P}} f(\mathbf{x},\mathbf{p})$
>
> where $f(\mathbf{x},\mathbf{p}) := \mathbf{p}^\top \mathbf{h}(\mathbf{x})$ by assuming that
> * $\tilde{\xi}$ is a discrete random variable with a finite support,
> * $\mathbf{h}(\mathbf{x})$ is the vector of realizations of $h(\mathbf{x}, \tilde{\xi})$,
> * and $\mathbf{p}$ is the vector of probability masses of realizations of $h(\mathbf{x}, \tilde{\xi})$.
>
> The above relationship only implies that DRO can be converted to a risk-averse optimization problem, but not the other way around. In particular, the above risk-averse optimization problem only concerns with the best-case (or worst-case) of $f(\mathbf{x},\mathbf{p})$ for $\mathbf{p} \in \mathcal{P}$ (via the term $\sup_{\mathbf{p} \in \mathcal{P}} f(\mathbf{x},\mathbf{p})$). On the other hand, our problem involves a risk level $\alpha$ that allows the flexibility in controlling the risk through $\alpha$ (via the CVaR at risk level $\alpha$ of $f(\mathbf{x}, \mathbf{W})$), which cannot be seen in the above formulation. In fact, the above formulation is only analogous to the adversarially robust BO [1] which is a special case of BO with VaR as shown in [12], i.e, BO with VaR (or CVaR) is more general than the above formulation (by replacing $\sup_{\mathbf{p} \in \mathcal{P}}$ with a risk measure such as VaR or CVaR that depends on an adjustable risk level $\alpha$). We will include this discussion in our revised paper to clarify this relationship.
>
>
>
> Regarding the selection of $\mathbf{w}_t$, as you noticed, we rely on lacing values (LV) and $\alpha_t$ to ensure the no-regret guarantee. We note that $\alpha_t$ still depends on the given $\alpha$.
> Selecting $\mathbf{w}_t$ without $\alpha_t$ will require a different theoretical analysis that we have not thought about. But we can give an example to show that selecting $\mathbf{w}_t$ that maximizes the uncertainty of the function evaluated at $\mathbf{x}_t$ (if we understand your suggestion correctly) may not be an efficient approach.
>
> Intuitively, the approach of maximizing the uncertainty of the function value may not be as efficient as selecting $\mathbf{w}_t$ as an LV because LV is constructed utilizing the structure behind VaR (and hence CVaR) while the uncertainty (e.g., posterior variance) is not much related to the definition of VaR (or CVaR). For example, let us consider the case that the distribution of $\mathbf{W}$ is defined as $P(\mathbf{W} = 0) = 0.1$, $P(\mathbf{W} = 1) = 0.2$, and $P(\mathbf{W} = 2) = 0.7$ and CVaR at risk level $\alpha=0.1$. Suppose at iteration $t$, the posterior beliefs of $f(\mathbf{x}_t, \mathbf{W})$ are:
>
> * $f(\mathbf{x}_t, \mathbf{W}=0) \sim \mathcal{N}(-100,1)$, i.e., a Gaussian distribution with mean $-100$ and variance $1$.
> * $f(\mathbf{x}_t, \mathbf{W}=1) \sim \mathcal{N}(0,10)$
> * $f(\mathbf{x}_t, \mathbf{W}=2) \sim \mathcal{N}(100,20)$
>
> From the above distributions, we can see that with a very high probability that $f(\mathbf{x}_t, \mathbf{W}=0) < f(\mathbf{x}_t, \mathbf{W}=1)$ and $f(\mathbf{x}_t, \mathbf{W}=0) < f(\mathbf{x}_t, \mathbf{W}=2)$. Furthermore, $P(\mathbf{W} = 0) = 0.1 = \alpha$. Thus, CVaR at $\alpha=0.1$ of $f(\mathbf{x}_t, \mathbf{W})$ is equal to $f(\mathbf{x}_t, \mathbf{W}=0)$ with a very high probability. However, if we select $\mathbf{w}_t$ as the one that maximizes the uncertainty of the function evaluated $\mathbf{x}_t$, we will select $\mathbf{w}_t = 2$ (the posterior variance is $20$) to obtain the observation of $f(\mathbf{x}_t, \mathbf{W}=2)$. This observation clearly does not help improving the uncertainty of CVaR of $f(\mathbf{x}_t, \mathbf{W})$ as this CVaR value is equal to $f(\mathbf{x}_t, \mathbf{W}=0)$.
>
>
>
> We also share your thought on the importance of the concept of lacing values. Therefore, in lines 192-199, we aim to explain the intuition behind the need for lacing values so that readers can grasp the concept of lacing values without going through its existence proof. We will try to briefly summarize the proof from [12] in our revised paper.
>
>
>
> Regarding the theoretical analysis when $\mathbf{W}$ is continuous, the main challenge is in obtaining (26) in Appendix F because $W_{l_{t-1}}^{\text{upper}}$ is an uncountable set (when $\mathbf{W}$ is a continuous random variable). For practitioners, we can approximate the continuous $\mathbf{W}$ with a discrete random variable on a finite support constructed from samples of $\mathbf{W}$ like in [4].
>
>
>
> To make our experiment section more exciting, we will consider a common situation in manufacturing processes: we would like to optimize certain product designs or materials while there exist some uncontrolled factors in the process such as friction and errors in the control inputs. In particular, we will include an experiment to reflect this interesting real-world application in our revised paper: *minimizing the residuary resistance per unit weight of displacement of a yacht through searching for the optimal hull geometry coefficients of the yacht in the face of the uncertainty in the Froude number* (the Froude number depends on the real-world environment and we assume that it can be simulated with computers during the optimization). The ground truth function is constructed using the yacht hydrodynamics data set (from the UCI machine learning repository). The optimization results we obtained using our proposed algorithms CV-UCB and CV-TS in comparison to $\rho \text{KG}^{\text{apx}}$ are shown in the following table. We observe that our proposed algorithms outperform $\rho \text{KG}^{\text{apx}}$ by finding better sets of hull geometry coefficients (which have lower regrets).
>
> | Algorithm                     | Regret &nbsp; &nbsp; &nbsp; &nbsp; &nbsp; &nbsp; &nbsp; &nbsp; &nbsp; &nbsp; &nbsp; |
> | ----------------------------- | ------------------- |
> | $\rho \text{KG}^{\text{apx}}$ | $1.5336 \pm 0.0032$ |
> | CV-UCB                        | $0.0777 \pm 0.0608$ |
> | CV-TS $k=3$                   | $0.0660 \pm 0.0675$ |
>
>
>
> We sincerely hope that the above clarifications and the additional experiment result will improve your opinion of our paper. We will address your comments about lines 162-166 and other minor issues in our revised paper.

---

### Official Review · Reviewer_ySFq · 2021-07-16

**Rating:** 7
**Confidence:** 3

**Summary:**

This paper proposes Bayesian Optimization (BO) approaches optimizing the conditional value-at-risk (CVaR) of the objective function distribution. Assuming the access to the objective function f(x, w) with full control of the environmental parameter w in the optimization time and assuming the prior distribution p(w) of the environmental parameter, the authors develop two algorithmic variants to optimize CVaR of  a given risk level.

The main contribution is the theoretical guarantee of the proposed variants. For the first variant, CV-UCB, the cumulative regret bound has been proved. For the second variant, CV-TS, the upper bound for the Bayesian cumulative regret has been proved.

The proposed approach has been tested on 3 synthetic benchmark problems and more realistic problems including portfolio optimization and simulated robot pushing experiment. The proposed approaches have been compared to an existing approach from [4], which can be applied to CVaR but not theoretical guarantee has been developed for. The results show competitive or superior performance.

**Limitations And Societal Impact:**

The limitation is not quite described. Please refer to the main review above.

**Main Review:**

This paper is very well written. This research topic is an important direction. The theoretical results are strong.

The only weakness of the paper is in experimental evaluation.
1. The tested problems are rather low dimensional problems, both in x and w. Especially for w, if I am not mistaken, the test problems are all of dimension no more than 2. I think this is a limitation. I recommend to add results on higher dimensional problems or to mention this limitation.
2. Though the authors list the computational cost  as one of the shortcomings of the approach in [4], I couldn't find any comment on the computational time in this experiments. It is advised to mention the computational time for each approach on each experiments.
3. It is interested to see how well the proposed approach works compared to other types of BO methods, such as the standard BO that simply optimize the expected value and the BO solving minimax optimization problems like [1], though they are not optimizing CVaR. It is often the case that minimizing the expectation contributes to minimizing the worst case, though not optimal. I strongly recommend to include the results of these baselines to enhance the empirical evaluation and revealing a possible limitation.

**Time Spent Reviewing:**

2

---

> ### Author Response · Authors · 2021-08-10
> **Author Response**
>
> We are very grateful for your acknowledgment of the important research direction and our strong theoretical results. In the following response, we will further clarify your comments on our experimental evaluation.
>
>
>
> 1. As you may have noticed, the dimension of $\mathbf{x}$ in the Hartmann-6D function is $5$. As $\mathbf{x}$ is the optimization variable, the problem will become harder for a larger dimension of $\mathbf{x}$ (we need to search for the optimal $\mathbf{x}$ in a larger space). This is the reason we focus on a moderate dimension of $\mathbf{x}$ in the experiment with Hartmann-6D. On the other hand, the dimension of $\mathbf{w}$ is linked with the difficulty in learning the value of CVaR. We have obtained a new experiment with Hartmann-6D where the dimension of $\mathbf{w}$ is set to $5$ to demonstrate the performance of our algorithm when the dimension of $\mathbf{w}$ is moderately large (in comparison to the existing method $\rho \text{KG}^{\text{apx}}$). As shown in the following table, we observe that our proposed algorithms (CV-UCB and CV-TS) still outperform the existing $\rho \text{KG}^{\text{apx}}$ by converging to lower regret values.
>
> | Algorithm                     | Regret &nbsp; &nbsp; &nbsp; &nbsp; &nbsp; &nbsp; &nbsp; &nbsp; &nbsp; &nbsp; &nbsp; |
> | ----------------------------- | ------------------- |
> | $\rho \text{KG}^{\text{apx}}$ | $0.1162 \pm 0.0923$ |
> | CV-UCB                        | $0.0088 \pm 0.0074$ |
> | CV-TS $k=3$                   | $0.0016 \pm 0.0025$ |
>
>
>
> 2. The claims about the computational efficiency in our paper are supported by the time complexity analysis in lines 162-166 and lines 178-183, which is more efficient than the time complexity analysed in Cakmak et al. 2020. We will restate the time complexity of Cakmak et al. 2020 in our revised paper to highlight the difference. This is also observed in our experiments, e.g., in the experiment with the Branin-Hoo function, given 45 observations, $\rho\text{KG}^{apx}$ takes 193.12 seconds on average to query, while our CV-UCB takes 3.53 seconds, and our CV-TS with a batchsize of 3 queries takes 23.38 seconds on average.
>
>
>
> 3. While we agree that standard BO may work for some experiments, the difference between the CVaR of the solution obtained with the standard BO that maximizes the expected value and the optimal CVaR can be very large. Let us consider an example where we want to maximize CVaR at risk level $\alpha=0.1$ of the function f(x,Z) = x*Z where $x \in [0,10]$, and Z is a discrete random variable specified by P(Z = -1) = 0.1, P(Z = 0) = 0.8, and p(Z = 10) = 0.1. We observe that the maximum CVaR is achieved at $x=0$ (CVaR is 0 at $x=0$ while CVaR =-x for $x > 0$). On the contrary, the maximum expected value $E_{Z} f(x,Z) = -0.1x + 0.1 \times 10x = 0.9x$ is achieved at $x=10$ where CVaR=-x is minimized. In other words, maximizing the expected value results in the worst possible CVaR value in this example. Regarding the solution to the minimax optimization problem in [1], we note that our CV-UCB also selects the same input queries as [1] when $\alpha \rightarrow 0^+$. We will include this discussion in our revised paper.

---

> > ### Comment · Reviewer_ySFq · 2021-09-01
> > **Thank you for your response**
> >
> > I appreciate your response. For the first point, though adding a result on a single problem is not enough to convince, I believe adding the new result will improve the paper. For the second point, thanks for the clarification. For the third point, I am suggesting to compare with the standard BO to see how much you gain by explicitly considering CVaR.

---

> > > ### Author Response · Authors · 2021-09-01
> > > **Thank you for your positive response and the acknowledgment of the computational efficiency; we like to give more experimental results and clarifications**
> > >
> > > Thank you for your response and acknowledgment of our claim about computational efficiency. We would like to provide additional clarification and experiments as follows:
> > >
> > > Regarding the first point, apart from the above new experimental results with Hartmann-6D, we also performed another experiment to assess the empirical performance of the proposed algorithms in higher input dimension than that in the main paper. We adopt the function $f_6(\mathbf{x}, \mathbf{w})$ from [4] where $\dim(\mathbb{X}) = 4$ and $\dim(\mathbb{W}) = 3$. The following results show that our algorithms (CV-UCB and CV-TS with $k=3$) outperform the existing $\rho \text{KG}^{\text{apx}}$.
> > >
> > > | Algorithm                     | Regret &nbsp; &nbsp; &nbsp; &nbsp; &nbsp; &nbsp; &nbsp; &nbsp; &nbsp; &nbsp; &nbsp; |
> > > | ----------------------------- | ------------------- |
> > > | $\rho \text{KG}^{\text{apx}}$ | $3.362 \pm 0.399$ |
> > > | CV-UCB                        | $0.008 \pm 0.005$ |
> > > | CV-TS $k=3$                   | $0.003 \pm 0.002$ |
> > >
> > > Regarding the third point, in the previous response, we showed a simple example where maximizing the expected value results in $\mathbf{x}$ that has the worst value of CVaR. Therefore, it suggests that using standard BO to optimize the expected value is not a good solution to optimize CVaR. Additionally, we optimized the expected value $E_Z f(x,Z)$ in the experiments with the Branin-Hoo and Goldstein-Price functions. The regrets (with respect to the optimal CVaR) achieved by optimizing the expected value $E_Z f(x,Z)$ in these 2 experiments are $0.3745$ and $0.4606$, respectively. These values are worse than the regrets achieved with CV-UCB, CV-TS, and $\rho \text{KG}^{\text{apx}}$ (BO that optimizes CVaR directly) which are at most $0.03$ for the experiment with Branin-Hoo and at most $0.09$ for the experiment with Goldstein-Price.
> > >
> > > Thank you again for your suggestion. We will include the above results and discussion in our revised paper.

---

### Official Review · Reviewer_3Pjv · 2021-07-16

**Rating:** 5
**Confidence:** 4

**Summary:**

This paper proposes to methods to maximize the conditional value-at-risk (CVaR) of a black-box function: the CV-UCB (Upper confidence bound) and the CV-TS (Thompson sampling)


**Main Review:**

- This paper is very relevant to reference [12], in which the Bayesian optimization algorithms for VaR has been considered. It is unclear to me why the CVaR objective may be *significantly* different or more difficult than the VaR objective. Both VaR and CVaR are risk measures, and from the theory side, one can use any other risk measures as an objective (spectral risk measure, distortion risk measure, etc.). The critical step in the UCB algorithm is to construct the confidence bound in equation (6), however, this bound can be easily generalizable as well by plugging in the corresponding formula of other risk measures. The regret bound also follows with minimal modification from Srinivas et al. (2010), and differs minimally from the VaR approach as in Lemma 3 of [12].

- The batch extension for CV-TS is straightforward for Thompson sampling which naturally handles parallel evaluations. This idea was also used previously in [12]

- The experimental results are limited and not convincing. No real-world application is considered, and the simulated robust pushing experiment does not look very promising.



**Time Spent Reviewing:**

7

---

> ### Author Response · Authors · 2021-08-10
> **Author Response**
>
> Thank you for your valuable feedback. We would like to address your comments and provide an additional experiment result which will be included in our revised paper.
>
>
>
> Although there are a large number of risk measures, we focus on a specific risk measure CVaR and perform a complete treatment of Bayesian optimization (BO) with CVaR (through CV-UCB and CV-TS with batch queries and theoretical analysis) so that the paper is easy to follow and coherent. CVaR is chosen because of its popularity and great interest to practitioners [18]. It is noted that existing BO works also focus on this risk measure (and also VaR) [4,21].
> The fact that we are the first to perform these theoretical analyses for CVaR and your comment that our bound may be "easily generalizable to other risk measures" imply that our work is an important stepping stone for BO with other risk measures. However, we also note that there exist risk measures, e.g., entropic value-at-risk and entropic risk measure, that are not immediately obvious (at least to us) to perform BO, which makes them future works. In brief, we believe our contribution to the important risk measure CVaR (by devising CV-UCB, CV-TS with batch queries) is substantial like existing BO works [4,12,21] that focus on VaR and CVaR.
>
> We motivate the use of CVaR over VaR as the objective function in lines 33-40 in the introduction with reference to [18], e.g., sensitive to the extreme tails.
> The difference between CVaR and VaR is further elaborated in Remark 1 (CVaR vs.VaR) and shown in Figure 1 (as acknowledged by Reviewer HzFE).
>
> The reasons why optimizing CVaR objective is difficult are described in lines 128-131 (the difficulty due to the distribution of CVaR), lines 142-147 (the difficulty relative to the existing solution of optimizing VaR [12]), and lines 247-251 (challenges to apply existing analysis of Thompson sampling algorithms).
>
>
>
> It is also noted there has not been any BO work with a performance guarantee for optimizing CVaR (with either 1-input query or batch queries). Therefore, we believe that our theoretical analyses *cannot* be obtained by *minimally modifying* existing works [12,20] (at least to us). For example, the following discussion highlights challenges in obtaining our analysis and its originality.
> * In the analysis of CV-TS, lines 249-251 explain the difficulty in obtaining the bounds of the tail expectation of the distribution of CVaR which is non-Gaussian and whose support is unbounded. We overcome this challenge through 3 steps of deriving the bound of the tail expectation of CVaR (lines 253-256). This technique is novel as it has not been applied to any Thompson sampling algorithms before to the best of our knowledge. Furthermore, the 3rd step is only made possible by identifying and proving an important property of VaR in Lemma 3 in our paper which has not been investigated before (e.g., it is not present in [12]). Thus, we respectfully disagree with you that the theoretical analysis can be obtained from [12] with minimal modification.
> * The selection of $\mathbf{w}_t$ can be a surprise to readers (as acknowledged by Reviewer HzFE) without looking at the analysis or our intuition in lines 188-191. This is because $\mathbf{w}_t$ is selected with respect to a risk level $\alpha_t$ different from the given risk level $\alpha$. We believe it reflects the distinction/novelty of our algorithms compared with other existing UCB and Thompsons sampling algorithms.
>
>
>
> We note that the batch extension for CV-TS is only possible given the challenging analysis of CV-TS is resolved (as explained above). Furthermore, while Thompson sampling naturally handles parallel evaluations, it is not as simple to analyse its regret (in Appendix G). If we understand your comment correctly, you have made a mistake in pointing out that this idea (i.e., Thompson sampling) was used in [12] previously. In fact, there is not any Thompson sampling algorithm or batch BO in [12].
> We note that Thompson sampling was used in [21] but (i) our CV-TS and their algorithm are different and (ii) there is not any theoretical analysis in [21].
> To the best of our knowledge, our analysis of the Thompson sampling algorithm for BO with CVaR is the first in the literature.
>
>
>
> We motivated our work through a real-world application in maximizing the crop yield in lines 20-26 in the introduction. In the experiments, we illustrate the performance on 2 real-world applications with simulated datasets like existing works [4,12].
> In the simulated robot pushing experiment, we can observe a clear difference between the performance of $\rho\text{KG}^{\text{apx}}$ and CV-TS k=3 (Figure 2e).
>
> We will include a new experiment constructed from a real-world dataset in our revised paper to enrich our experiment section. This experiment reflects a potential and important application of our solution in manufacturing processes: we would like to optimize certain product designs or materials given the uncertainty in some uncontrolled factors in the process such as friction and errors in the control inputs. In particular, we will include an experiment about *minimizing the residuary resistance per unit weight of displacement of a yacht through searching for the optimal hull geometry coefficients of the yacht in the face of the uncertainty in the Froude number* (the Froude number depends on the real-world environment and we assume that it can be simulated with computers during the optimization). The ground truth function is constructed using the yacht hydrodynamics data set (from the UCI machine learning repository). The optimization results we obtained using our proposed algorithms (CV-UCB and CV-TS) and the existing $\rho \text{KG}^{\text{apx}}$ are shown in the following table. We observe that our proposed algorithms outperform $\rho \text{KG}^{\text{apx}}$ by finding better sets of hull geometry coefficients (which have lower regrets).
>
> | Algorithm                     | Regret &nbsp; &nbsp; &nbsp; &nbsp; &nbsp; &nbsp; &nbsp; &nbsp; &nbsp; &nbsp; &nbsp; |
> | ----------------------------- | ------------------- |
> | $\rho \text{KG}^{\text{apx}}$ | $1.5336 \pm 0.0032$ |
> | CV-UCB                        | $0.0777 \pm 0.0608$ |
> | CV-TS $k=3$                   | $0.0660 \pm 0.0675$ |
>
>
>
> We sincerely hope that the above justifications and experiment results will improve your perception of our paper.

---

> > ### Comment · Reviewer_3Pjv · 2021-08-22
> > **Score raised to 5**
> >
> > I thank the authors for the clarifications which have partially resolved some of my concerns. I would recommend the authors to include an in-depth discussion about the difference against reference [12] in the revision.

---

> > > ### Author Response · Authors · 2021-08-23
> > > **Thank you for your positive response; Elaboration on our discussion in the paper about the key differences with [12]**
> > >
> > > Thank you for your positive response. We are glad that our clarifications resolved your concerns. We like to elaborate on our existing discussion in the paper about the key differences with reference [12]:
> > >
> > >
> > > **Difference in problem definition between CVaR (our work) and VaR [12]**
> > >
> > > CVaR is sensitive to values at the extreme tails of the distribution of $f(\mathbf{x},\mathbf{W})$ while VaR is not (lines 33-36). For example, consider a portfolio allocation problem: CVaR can identify the risk of the worst-case value of the return (by being sensitive to the values at the extreme tail) while VaR cannot (lines 36-40).
> > >
> > > Remark 1 (CVaR vs. VaR) is dedicated to explaining the visualization in Figure 1 about the difference between CVaR and VaR: Figure 1 shows 2 distributions with the same VaR but different CVaR. The difference in CVaR values is due to the difference in the tails of the two distributions, which has an implication on the risk (lines 97-98). We also connect the argument in this remark back to the portfolio optimization problem in lines 98-101 in Remark 1.
> > >
> > > **Difference in solution between CVaR (our work) and VaR [12]**
> > >
> > > Novel UCB-based approach: CVaR is defined as the expectation of function values that are at most VaR at the same risk level [4]. Thus, optimizing CVaR involves learning the probabilities of function values less than VaR, which cannot directly utilize the solution of [12] (that optimizes VaR). This point was explained in lines 142-147 in our paper.
> > >
> > > Novel Thompson sampling approach and novel batch BO: There is no mention of the Thompson sampling or batch queries in [12], which consequently implies the significance and novelty of our proposed algorithm and theoretical analysis.
> > >
> > > We hope that the above detailed comparison in our paper is what you are looking for. Please let us know if you think any other details can be included to better clarify the comparison with [12]. Thank you.

---

### Official Review · Reviewer_sLCu · 2021-07-25

**Rating:** 7
**Confidence:** 4

**Summary:**

This paper considers Bayesian optimization (BO) of objective functions of the form $c_f(x;\alpha) = \mathrm{CVaR}_\alpha(f(x, W))$, where $f$ is modeled as a Gaussian process (GP),, and $\mathrm{CVaR}_\alpha$ denotes the $\alpha$-level conditional value-at-risk, which is computed with respect to the randomness over $W$. This work focuses on the simulation optimization setting, where it is possible to evaluate $f(x,w)$ for each $(x,w)\in\mathbb{X}\times\mathbb{W}$, and develops two acquisition functions to select the pairs $(x,w)$ at which to evaluate $f$, CV-UCB and CV-TS, which extend the classical UCB and Thompson sampling acquisition functions, respectively. These algorithms select the $x$ component of the pair and are complemented with a common strategy to select $w$. Given $x_t$, the $x$ component of the pair to be evaluated at time $t$, this strategy first chooses $\alpha_t$ that maximizes the difference of the VaR of an upper confidence bound on $f(x_t, W)$, $u_t(x, W)$, and the VaR of a lower confidence bound on $f(x_t, W)$, $l_t(x, W)$. Finally, given $x_t$ and $\alpha_t$, this strategy chooses $w_t$ so that   $[\mathrm{VaR}_\alpha (l_t(x, W)),  \mathrm{VaR}_\alpha (u_t(x, W))] \subset [l_t(x_t, w_t), u_t(x_t, w_t)]$ (for $\alpha= \alpha_t$). A $w_t$ satisfying this condition is called a lacing value. The two resulting algorithms are shown to enjoy sublinear cumulative regrets when $\mathbb{X}$ and $\mathbb{W}$ are both finite. Finally, the empirical performance of these algorithms is evaluated in three synthetic experiments and two real-world experiments, showing favorable results.

**Limitations And Societal Impact:**

The authors have implicitly discussed the limitations of their work. However, I it would be beneficial to summarize them and discuss them more explicitly in the conclusion. The societal impact of this work has not been discussed.

**Main Review:**

Originality: The proposed algorithms are, to the best of my knowledge, novel, and are the first ones for GP-based optimization of CVaR that come with theoretical guarantees. However, these algorithms (including their theoretical analysis) heavily rely on the work of Nguyen et al. 2021. The algorithmic contributions of this paper are, therefore, somewhat limited. In this regard, the most novel aspect of this work is the CV-TS algorithm, which allows for batch evaluations. In addition, as acknowledged by the authors, the problem setup has been previously considered in the literature (see, e.g., Cakmak et al. 2020), and the test problems are the same (translated from VaR to CVaR) as those in Nguyen et al. 2021. As a consequence, the overall novelty of this work is limited.

Quality: The proposed algorithms are technically sound. The proofs on the regret bounds seem to be correct (although did not check in detail some of the calculations).  With only three synthetic test problems and two realistic test problems, the empirical evaluation is somewhat limited. Moreover, claims about the computational efficiency of the proposed algorithms have not been properly backed up. For this, I would suggest the authors to include a table of the average running times of each algorithm in each problem.

Clarity: This paper is fairly well written and easy to follow. One suggestion is that the paper would benefit from a picture illustrating the behavior of the proposed algorithms.

Significance: Risk-averse Bayesian optimization is a problem of high practical relevance. Moreover, the only existing algorithms (for the CVaR setting), $\rho$KG and its approximate variant, are fairly computationally expensive. Therefore, cheaper algorithms with good empirical performance are relevant and prone to be adopted by practitioners. The algorithms proposed in this paper seem to poses these characteristics. However, a more thorough empirical evaluation is required.

**Time Spent Reviewing:**

6 hours

---

> ### Author Response · Authors · 2021-08-10
> **Author Response**
>
> We very much appreciate your valuable feedbacks and your acknowlegement of the need of cheaper algorithms with good empirical performance for practitioners such as our proposed algorithms. In the following response, we hope that your questions on the originality aspect of our work and the computational complexity can be resolved. Furthermore, we provide an additional experiment result with a real-world dataset to further strengthen our empirical advantage.
>
>
>
> While our work relies on the notion of lacing values which is introduced in the context of VaR in Nguyen et al. 2021, our work is the first work that handles both 1-input query and a batch query with performance guarantee in this problem setup (optimizing CVaR of a black-box function with input queries $(\mathbf{x}_t, \mathbf{w}_t)$) in comparison to Cakmak et al. 2020 and Nguyen et al. 2021 as you have noticed. Furthermore, it requires a notable amount of work to construct the algorithms and perform the theoretical analysis for the following reasons.
>
> First, in the analysis of CV-TS, we need to overcome the challenge in the distribution of CVaR (it is non-Gaussian and its support is unbounded) (in lines 249-251). This problem is challenging (at least to us) as it requires 3 steps of deriving the bound of the tail expectation of CVaR (lines 253-256). This technique is original as it has not been applied to any Thompson sampling algorithm before to the best of our knowledge.
>
> Second, a novel property of VaR discovered in our work is described in Lemma 3 which has not been explored in the work of Nguyen et al. 2021.
>
> Third, for both CV-UCB and CV-TS, the selection of $\mathbf{w}_t$ can be a surprise to the reader (as acknowledged by Reviewer HzFE) without looking at the analysis or our intuition in lines 188-191. This is because $\mathbf{w}_t$ is selected with respect to a risk level $\alpha_t$ different from the given risk level $\alpha$. We believe it reflects the originality in our algorithms compared with existing UCB and Thompson sampling algorithms where such component does not exist.
>
>
>
> Although our experiments are taken from existing works in the literature, our purpose is different: we would like to demonstrate that our proposed approaches are able to optimize CVaR of a black-box function with both 1-input query and a batch query.
>
> We will include a new experiment constructed from a real-world dataset in our revised paper to enrich our experiment section. This experiment reflects a potential and important application of our solution in manufacturing processes: we would like to optimize certain product designs or materials given the uncertainty in some uncontrolled factors in the process such as friction and errors in the control inputs. In particular, we will include an experiment about *minimizing the residuary resistance per unit weight of displacement of a yacht through searching for the optimal hull geometry coefficients of the yacht in the face of the uncertainty in the Froude number* (the Froude number depends on the real-world environment and we assume that it can be simulated with computers during the optimization). The ground truth function is constructed using the yacht hydrodynamics data set (from the UCI machine learning repository). The optimization results we obtained using our proposed algorithms (CV-UCB and CV-TS) and the existing $\rho \text{KG}^{\text{apx}}$ are shown in the following table. We observe that our proposed algorithms outperform $\rho \text{KG}^{\text{apx}}$ by finding better sets of hull geometry coefficients (which have lower regrets).
>
> | Algorithm                     | Regret &nbsp; &nbsp; &nbsp; &nbsp; &nbsp; &nbsp; &nbsp; &nbsp; &nbsp; &nbsp; &nbsp; |
> | ----------------------------- | ------------------- |
> | $\rho \text{KG}^{\text{apx}}$ | $1.5336 \pm 0.0032$ |
> | CV-UCB                        | $0.0777 \pm 0.0608$ |
> | CV-TS $k=3$                   | $0.0660 \pm 0.0675$ |
>
>
>
> Our claim regarding the computational efficiency is backed up by the time complexity of our algorithms shown in lines 162-166 and lines 178-183, which is more efficient than the time complexity analysed in Cakmak et al. 2020. We will restate the time complexity of Cakmak et al. 2020 in our revised paper to highlight the difference. This is also observed in our experiments, e.g., in the experiment with the Branin-Hoo function, given 45 observations, $\rho\text{KG}^{apx}$ takes 193.12 seconds on average to query, while our CV-UCB takes 3.53 seconds, and our CV-TS with a batchsize of 3 queries takes 23.38 seconds on average.
>
>
>
> We sincerely hope that the above clarifications and the additional experiment results will improve your opinion of our paper. We will address your other comments (such as explicitly discussing the limitations in the conclusion) in our revised paper.

---

> > ### Comment · Reviewer_sLCu · 2021-08-22
> > **Thank you for your response and additional requests**
> >
> > De authors,
> >
> > Thank you for your response. Some of my concerns have been resolved. In particular, the novelty in your technical contributions with respect to the work Nguyen et al. 2021 is now more evident to me. I also appreciate that you are planning to include a new realistic experiment and have provided promising preliminary results. However, I still think that your empirical evaluation should be improved. I would like to request the following:
> >
> > 1. I noticed that $\rho$KG$^\mathrm{apx}$ is not among the algorithms included in your code. I am thus assuming that you are using the implementation provided by Cakmak et al. 2020. For a fairer comparison, it would be better if you re-implemented this algorithm. I understand this might be challenging so I would like you to at least explain the steps you have been taken to guarantee that this is a fair comparison (e.g., are the GP priors used by CV-UCB, CV-TS, and  $\rho$KG$^\mathrm{apx}$ the same?).
> > 2. Could you please include the average runtimes for all your test problems here?
> > 3. Like reviewer ySFq, I see the low-dimensionality of all the test problems (in both $x$ and $w$, but in particular in $w$) as a weakness of your empirical evaluation. It would be desirable to have more experiments where $\mathrm{dim}(\mathbb{X})\geq 2$ and $\mathrm{dim}(\mathbb{W})\geq 3$ (simultaneously). I noticed in your response to ySFq that you ran one variation of the Hartmann-6D problem with $\mathrm{dim}(\mathbb{W})=5$, but a single experiment (in particular one that is simply a variation of one of your current experiments) does not sufficiently address this concern.

---

> > > ### Author Response · Authors · 2021-08-22
> > > **Thank you for your recognition of the technical novelty and our new realistic experiment; we like to present another experiment with a higher input dimension**
> > >
> > > Thank you a lot for acknowledging that we have resolved your concerns regarding the novelty in our technical contributions and your appreciation of our new realistic experiment.
> > >
> > > We will incorporate your helpful suggestion into our revised experiment section. Due to the time constraint, let us address some of your questions first.
> > >
> > > 3.We just ran our CV-UCB to optimize CVaR at risk level $\alpha=0.25$ of the function $f_6(x_c, x_e)$ in [4] where $\dim(\mathbb{W}) = 3$ and $\dim(\mathbb{X}) = 4$. With $80$ random initial data points, CV-UCB is able to improve the instantaneous regret from $19.993 \pm 3.461$ (at iteration $0$) to $0.008 \pm 0.005$ (at iteration $50$) over $3$ random experiments. We will continue running the code to obtain the results of $10$ random experiments and re-run $\rho \text{KG}^{apx}$ in this experiment for comparison.
> > >
> > > 1.It is indeed challenging to re-implement $\rho\text{KG}^{apx}$ as you commented, so we used the implementation by Cakmak et al. 2020 [4]. We will include more details to show that the comparison between our algorithms and $\rho\text{KG}^{apx}$ is fair. For example, we copied the prior on the noise variance from [4]: it is a Gamma distribution with the shape and rate parameters as $1.1$ and $1/0.5$, respectively (in the `fit_gp` function in `BoRisk/experiment.py` file in the code of [4]); in our code, as we used GPflow where the Gamma prior is defined with the shape and scale parameters (i.e., $1 / $ the rate parameter), the same prior is set for the noise variance as seen in the function `fit_gp` in `functions.py` in our submitted code. Another example is that as the initial data points are randomly generated, we also ensured that the same set of initial data points are used for both $\rho\text{KG}^{apx}$ and our algorithms at every random experiment. It can be seen from our submitted `ucb.py`, `ts.py`,  and scripts that the same random seed is used to generate the initial observations. These initial data points are also saved into a file (e.g., lines 338-391 in `ucb.py`) which is fed into code of $\rho \text{KG}^{apx}$ to ensure the same initialization. We will include other details in our revised paper.
> > >
> > > 2.We continue running the code to report the running time of our algorithms and $\rho\text{KG}^{apx}$. It takes some time due to the fact that we need to run experiments sequentially (running many experiments in parallel in the same machine will affect the measure of the running time). However, we would like to highlight that our time complexity (mentioned in our paper) is significantly better than that of $\rho\text{KG}^{apx}$ (in Appendix 1 in [4]). One clear difference is that the time complexity of $\rho\text{KG}^{apx}$ increases linearly with the number of fantasy models (samples of the function $f$ from the GP posterior belief) and the number of fantasy models required can be very large: Proposition 1 in [4] says that it requires the number of fantasy models approaches infinity in order to ensure an asymptotically unbiased and consistent estimator of the gradient. On the other hand, our CV-UCB does not involved any fantasy models and CV-TS requires the same number of fantasy models as the size of the batch queries.
> > >
> > > We truly hope that by incorporating our above explanations, especially the new experiment with $\dim(\mathbb{W}) = 3$ and $\dim(\mathbb{X}) = 4$, your opinion on our paper could be improved.

---

> > > > ### Comment · Reviewer_sLCu · 2021-08-22
> > > > **Thank you for your response and a remark on the kernel function**
> > > >
> > > > Dear authors,
> > > >
> > > > Thank you for your quick response. I look forward to seeing the results of your new experiments.
> > > >
> > > > Regarding the fairness of the comparison with $\rho\mathrm{KG}^\mathrm{apx}$, I was going through your work's supplementary material again and noticed that CV-UCB and CV-TS use GP models with a squared exponential kernel, whereas the implementation of Cakmak et al. 2020 seems to use a Matern kernel. Since the choice of the kernel plays a significant role in the performance of a BO algorithm, all acquisition functions should use the same kernel in order for the comparison to be fair. I am not sure if it is realistic to fix this before the final decision needs to be made, but please take this into consideration for a future version of your work.

---

> > > > > ### Author Response · Authors · 2021-08-24
> > > > > **Thank you for your further comment; and the experimental results for CV-TS and $\rho\text{KG}^{apx}$**
> > > > >
> > > > >
> > > > > We have obtained the results for CV-TS and the baseline $\rho\text{KG}^{apx}$ in the new experiment where $\dim(\mathbb{W}) = 3$ and $\dim(\mathbb{X}) = 4$. In the following table, both CV-UCB and CV-TS outperform $\rho \text{KG}^{apx}$. Besides, $\rho \text{KG}^{apx}$ also incurs a larger runtime. For example, at the $30$-th iteration, while CV-UCB and CV-TS (with $k=3$) only require, respectively, $47.78$ and $193.86$ seconds to optimize for the input query, respectively, $\rho\text{KG}^{apx}$ requires $460.37$ seconds (although due to the time constraint, we reduced the runtime of $\rho\text{KG}^{apx}$ by trading off with  the number of fantasy models and using only $80$ initial data points).
> > > > >
> > > > > | Algorithm                     | Regret &nbsp; &nbsp; &nbsp; &nbsp; &nbsp; &nbsp; &nbsp; &nbsp; &nbsp; &nbsp; &nbsp; |
> > > > > | ----------------------------- | ------------------- |
> > > > > | $\rho \text{KG}^{\text{apx}}$ | $3.362 \pm 0.399$ |
> > > > > | CV-UCB                        | $0.008 \pm 0.005$ |
> > > > > | CV-TS $k=3$                   | $0.003 \pm 0.002$ |
> > > > >
> > > > > Thank you for making us aware of the difference in the kernel. We will take note of this in our revised paper by incorporating experimental results with the same kernel. As you may have already known, SE is the Matern kernel when $\nu \mapsto \infty$ and the difference between Matern with large values of $\nu$ and SE is small, e.g., Matern with $\nu=2$ and SE are similar in Figure 4.1 of the Gaussian processes for machine learning book (page 85 in Rasmussen and Williams (2006)). Therefore, there may not be a big performance difference between using the Matern with $\nu=5/2$ in [4] and using an SE kernel. We will verify the performance gap using the same kernel in our revised paper.

---

> > > > > > ### Comment · Reviewer_sLCu · 2021-09-04
> > > > > > **Final thoughts on this paper**
> > > > > >
> > > > > > Dear authors,
> > > > > >
> > > > > > Thank you for conducting this new experiment. I still think that your empirical evaluation requires some polishing but, after this follow-up, I believe it is realistic for you to make several improvements on time in case your paper is accepted. As a consequence, I have decided to increase my score slightly.  Please take into account my suggestions and concerns in the final version of your work. In particular, please use the same kernel for all methods, and also consider conducting a few additional experiments. Finally, your algorithms seem to achieve state-of-the-art performance, but they are not straightforward to implement. Please consider releasing your code and test problems once your paper is accepted.
> > > > > >
> > > > > > Best wishes,
> > > > > >
> > > > > > Reviewer sLCu

---

> > > > > > > ### Author Response · Authors · 2021-09-04
> > > > > > > **Warmest thanks**
> > > > > > >
> > > > > > > Thank you for your time and effort in providing us constructive responses and helpful suggestions. We will consider them seriously in the revised paper and release the code.

---

### Decision · Program_Chairs · 2021-09-27

**Decision:**

Accept (Poster)

**Comment:**

This paper addresses Bayesian optimization to search a maximizer of the CVaR of the black-box objective function. Two variants inspired by UCB and Thompson sampling are presented. For CV-UCB, the cumulative regret bound has been proved and for CV-TS, the upper bound for the Bayesian cumulative regret has been proved. The paper is well written and the topic is interesting. A few reviewers pointed out the weakness of empirical evaluations. The authors promised to include an additional experiment whose preliminary results are reported in the rebuttal.The rebuttal was not fully satisfactory but the efforts in providing preliminary results were appreciated. During the committee discussion period, two of reviewers raised the score. Some concerns in the limited novelty over previous work still remains. However, the paper presents a solid theoretical guarantee, which might not be obviously derived from known results.